# Paradise fish (*Macropodus opercularis*) as a complementary translational model for emotional and cognitive function
Zoltán K. Varga [1] ✉, Diána Pejtsik[1,2], Tímea Csorvási[1,3], Éva Mikics [1], Ádám Miklósi[4] & Máté Varga [3] ✉

Zebrafish have revolutionised physiological screening in vertebrates, however, their strong sociality present challenges for interpreting behavioural assays conducted on individual subjects. To retain the advantages of the zebrafish model while addressing its limitations, we propose the use of a solitary species—the paradise fish—as a complementary model system. We compared paradise fish and zebrafish of late larval stage in social and non-social exploratory tasks, anxiety tests and in a working memory assay to assess their performance in these individual-based challenges. We found that in contrast to zebrafish, paradise fish did not show social approach in sociability tests, their exploratory behaviour was unaffected by the presence of a conspecific, and social isolation did not impair their performance during anxiety tests. Intra- and intertest variability measures of different anxiety tests revealed that, compared to zebrafish, paradise fish express more consistent, repeatable patterns of exploratory and risk-avoidance behaviour across time and contexts. We also showed that paradise fish exploration of the Y-maze is dominated by arm alternations, suggesting advanced working memory. Considering the results of this systematic comparison and the natural history of the two species we recommend prioritizing zebrafish in social tasks, while favouring paradise fish in individual-based behavioural assays.

Animal models are indispensable for biological research, especially to understand the mechanisms and causative relationships of biological processes. Different fields have benefited from a wide range of model species, each bringing with them specific trade-offs between simplicity versus similarity to humans[1–5]. Fish are positioned towards the middle of such imaginary scales, manifesting the hallmark features of a conserved vertebrate body plan, yet in a relatively simple form. While over the past decades different scientific fields favoured different fish species[6–9], zebrafish (*Danio rerio*) have lately become the predominant model-of-choice for biomedical sciences[10,11]. The popularity of zebrafish, however, is partly due to serendipity: it became one of the dominant models of genetics and as genetic tools started to dominate life sciences, zebrafish research found new niches.

Nowadays high-throughput and high-resolution screening can be combined with an ever expanding genetic toolkit, making zebrafish an extremely powerful model species, almost unparalleled among other vertebrates[12–14]. Nowhere else is this more apparent than in neuroscience: fish models let us assess neuronal activity simultaneously and continuously in the whole brain and on the level of single cells, in living animals[14]. These in vivo imaging methods combined with cell-specific manipulation techniques will elevate our knowledge about brain mechanism and function to a whole new level.

Despite these advances, the validity and generalisability of zebrafish behaviour remains to be assessed in this promising model system. In their natural habitat, zebrafish only exist in shoals[15], and their behavioural repertoire has evolved to function within social groups. As a result, their individual behaviour is determined by – and can only be interpreted within – this social context. Internal states and behavioural responses of zebrafish are highly affected by the presence or absence of conspecifics[16–23]. Mechanisms such as social buffering and social contagion tend to homogenise behaviour within the group, making the group itself a more appropriate unit of analysis[21,22]. Therefore, group-level assessment of behaviour cannot be correlated with individual physiological measures. Conversely, separating zebrafish from their shoal to assess individual behaviour constitutes a major and specific perturbation to both their external and internal

¹Translational Behavioral Neuroscience Research Group, HUN-REN Institute of Experimental Medicine, Budapest, Hungary. ²Semmelweis University Doctoral School, Budapest, Hungary. ³Department of Genetics, ELTE Eötvös Loránd University, Budapest, Hungary. ⁴Department of Ethology, ELTE Eötvös Loránd University, Budapest, Hungary. ✉e-mail: interzapp.varga@gmail.com; mvarga@ttk.elte.hu

states[23–26]. In either case, behavioural outcomes of the observed individuals in tests that designed to measure emotional or cognitive functions are likely to be at least partially biased. This raises a critical question: to what extent is the performance of an isolated individual representative of a species that has evolved to function within a group?

Here, we propose the use of an alternative fish species, the paradise fish (*Macropodus opercularis*) to keep the advantages and compensate for the limitations of the zebrafish model. A reference genome and details of the effective housing of paradise fish were recently published[27,28], while its behavioural repertoire was already well-described, including complex risk-avoidance[29–31], social[32,33] and cognitive responses[34,35]. The social dynamics of paradise fish, being a facultative social species, resembles more closely human social dynamics than that of zebrafish. Unlike zebrafish, which live in shoals, paradise fish exist in less cohesive social groupings, more comparable to human social settings. Paradise fish show a wide and context-dependent repertoire of social behaviours, but do not rely on constant group cohesion and primarily function as solitary individuals. Indeed, during nest building and the territorial defensive behaviour, paradise fish males act as solitaires. Males can also express agonistic interactions, such as during courtship behaviour and even more prominently during parental care (mouthing, bubbling, and nest maintenance), which can take up to 5-6 days[36]. Larval paradise fish are not strictly solitary and the transition towards solitary behaviour is a gradual process, with the first signs of increased aggression appearing in the juvenile stage and becoming more pronounced during the adolescent stage.

We hypothesise that a fish naturally adapted to cope with challenges alone is likely to exhibit more consistent behavioural responses when tested in individual-based tasks.

In this study, we aimed to compare the performance of zebrafish and paradise fish in individual-based behavioural assays, and to determine how their developmental social environment influence or bias their behavioural outcomes. We utilised fish of late larval stage (30 dpf), a period that combines the advantages of the larval and adult age: high-throughput or whole-brain screenings and decent performance in cognitive or social tasks. We assessed i) the sociability (approaching conspecifics), with or without the presence of the other species, and ii) the effect of conspecifics on exploratory behaviour and anxiety. Furthermore, we compared the performance of individual zebrafish and paradise fish in tasks modelling higher order processes, such as iii) anxiety-like behaviour and responses to anxiolytics, iv) behavioural consistency and v) working memory. We found that compared to zebrafish, paradise fish are less biased by extrinsic social and non-social factors and show more consistent and efficient performance in paradigms that model phenomena such as anxiety or working memory. We conclude that paradise fish is a more reliable model to be used in individual-based assays while zebrafish should be the model of choice in tasks that involve social challenges.

## Results

In the first three experiments we aimed to compare how the presence or absence of a conspecific affects the behaviour of the two species.

### Experiment 1: paradise fish show less direct social interactions compared to zebrafish

In our first experiment, we aimed to compare the sociability of zebrafish and paradise fish of late larval stage (30 dpf) in the U-shaped sociability test[18] in baseline or perturbed conditions. As a baseline, we measured the preference towards a single stimulus conspecific (intra-species challenge). As a control we also measured the preference towards the other species per se (inter-species challenge). To model perturbed conditions, the other species were also presented as stimulus (double-species challenge) (Fig. 1A).

While zebrafish showed similar swimming velocity in all contexts, paradise fish enhanced their swimming speed in inter- and double-species challenges, indicating a response to the presence of the zebrafish stimulus (Fig. 1B). These differences in locomotion between challenge types did not

appear in intersection entries (Fig. 1C). Interestingly, the comparison of the two species revealed that while paradise fish generally swim slower in the test, they visit the intersection zone more. In line with previous data, zebrafish preferred the proximity of a conspecific compared to an empty zone (Fig. 1C left). Interestingly, the presence of a stimulus paradise fish per se did not trigger the approach of zebrafish in the inter-species challenge, but made their social approach less pronounced in the double-species challenge, indicated by the only marginally significant conspecific preference of zebrafish in the latter condition. In contrast to zebrafish, paradise fish did not show preference towards a conspecific in either condition (Fig. 1C right): they spent approximately the same amount of time in both zones in the intra-species challenge, avoided zebrafish in the inter-species challenge, and avoided conspecifics in the double-species challenge. Time percent spent in each zone (Fig. 1D) indicate that both species spent most of their time either in the social or the non-social zone, but not in the intersection zone. In contrast, enter frequencies to the intersection zone (Fig. 1E) were higher compared to entries to the other two zones in all challenge types in paradise fish but not in zebrafish, indicating that paradise fish switch more between social and non-social zones. In summary, zebrafish, but not paradise fish, exhibited social approach behaviour in response to a conspecific. Conversely, paradise fish, but not zebrafish, showed avoidance behaviour when exposed to the other species. In the choice condition, zebrafish displayed a trend toward preferring the conspecific, whereas paradise fish showed a greater tendency to approach the heterospecific individual. All statistical parameters of the experiment are shown in Supplementary Table 1.

### Experiment 2: exploration of paradise fish is not biased by the presence of a conspecific

The stress-ameliorating role of conspecifics in social species is known as social buffering, a phenomenon that has been described in a wide range of species[30,37–42], including zebrafish[22]. Given the divergent responses to conspecifics in our first experiment next, we aimed to investigate social buffering on exploration of a novel environment. One or two individuals of zebrafish or paradise fish of late larval stage (30 dpf) were exposed to a 12-compartment linear maze called the slalom test (Fig. 2A) and mean transition latency and overall exploration success were measured (Fig. 2B,C). Zebrafish of early larval stage (8 dpf), which are known to show less discrete social activity compared to late larval individuals were also present as control. We found that 30 dpf zebrafish with a companion more actively explored the area than groups of 8 dpf conspecifics or paradise fish of the same age (30 dpf), indicated by the decreased mean transition latencies (Fig. 2B). Interestingly, larval and juvenile zebrafish show a bimodal exploration pattern in different conditions; alone or with a companion, respectively. Bimodality is apparent in larvae when those explore alone, while bimodality is presented in juveniles when those explore as a group of two. In contrast, paradise fish show a unimodal gaussian distribution of exploration of the novel area in every condition (Fig. 2D left). All distribution types were confirmed using gaussian mixed modelling (Fig. 2D right): the highest Bayesian information criterion (BIC) absolute values for one peak (an indication of unimodal distribution) were at assays that assessed the exploration of one 30 dpf zebrafish or of either one or two 30 dpf paradise fish. Interestingly, besides less dynamic exploration, paradise fish showed the highest success rate in exploring the whole area, irrespective of the group size (Fig. 2C). Thus, the exploration of paradise fish is more effective compared to zebrafish and is not biased by the current presence of a conspecific. All statistical parameters of the experiment are shown in Supplementary Table 2.

### Experiment 3: exploration of paradise fish is not biased by sub-chronic social isolation

Next, we aimed to investigate whether the subchronic (1–3 days) absence of conspecifics affects the behaviour of paradise fish. We introduced zebrafish or paradise fish of late larval stage (27–30 dpf) to social isolation on day 0

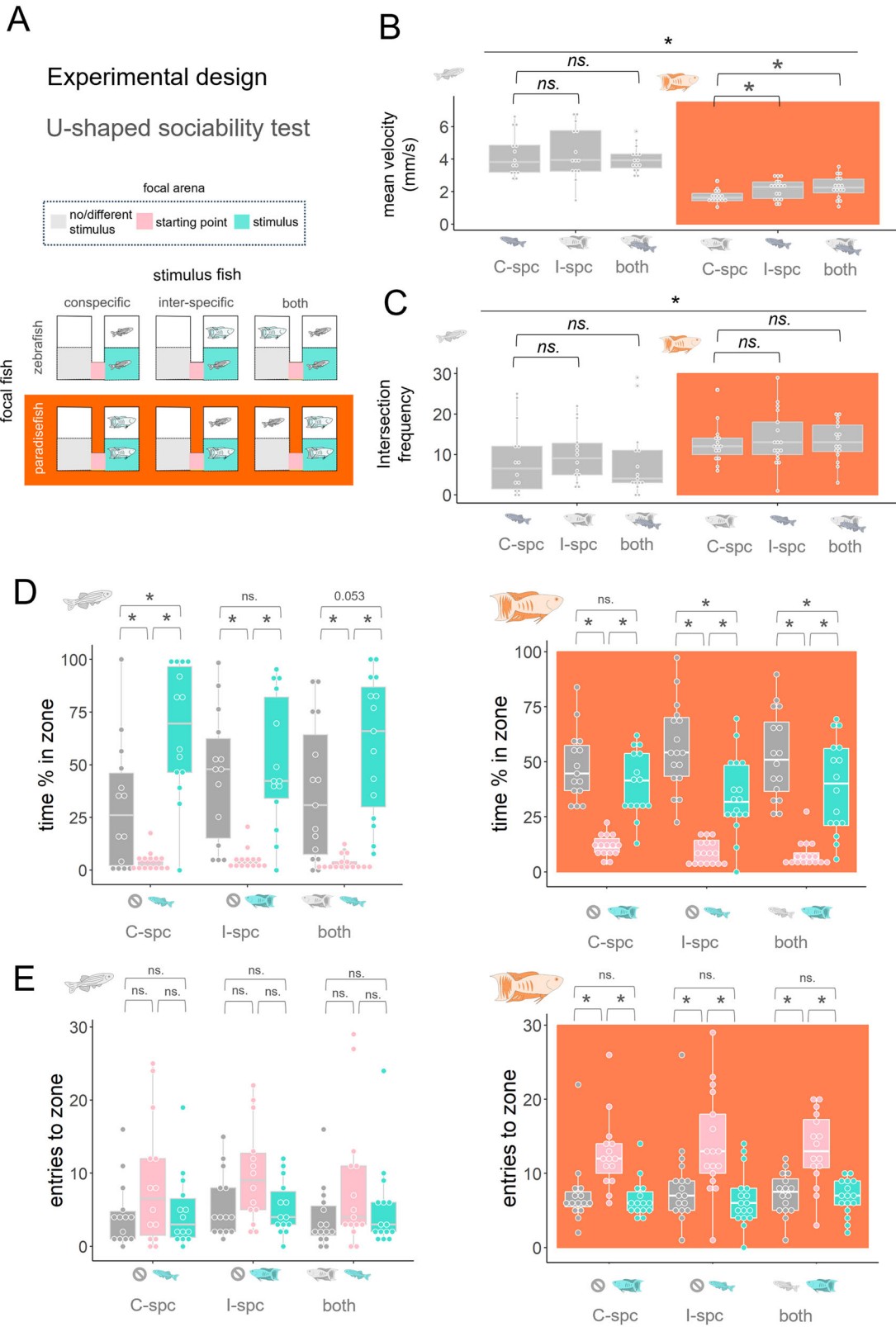

and tested their anxiety-like responses on day 1, 2 and 3 in the swimming plus-maze (SPM) and showjump (SJ) tests (Fig. 3A). Since social isolation affects both body size and locomotion in various species, we measured body length and centrum entries in both species and conditions. Paradise fish were smaller compared to zebrafish and made less visits to the centrum of the swimming plus-maze regardless of rearing conditions (Fig. 3B, C,

Supplementary Fig. 1B–D). We found that socially isolated zebrafish showed increased exploration latencies (Fig. 3D–G, left panels) and overall decreased exploration (Fig. 3D–G, right panels) in contrast to socially housed (tested or test-naive) individuals in both tests, an effect similar to that of chronic isolation[23,26]. Interestingly, despite both tests being based on the conflict between exploration of novel areas and the aversive proximity of

**Fig. 1 | Paradise fish show less direct social interactions compared to zebrafish.** **A** Experimental design of the intra-, inter-, and double-species challenges in which a conspecific (C-spc), an intraspecific (I-spc) or both are presented, respectively. 30 dpf focal subjects were put to the starting point/intersection (pink) of the focal arenas consisted of a no/different stimulus (grey) and a stimulus zones (turquoise). Following a fifteen-minute habituation period, stimulus subjects were placed next to the stimulus zones or next to the stimulus and the no/different stimulus zones in case of the double species challenge. **B** Mean velocity of swimming in the whole apparatus in response to each challenge presenting different stimulus fish. * on a solid line represents significant main effect of the species, * on bracket represent significant difference from the swimming velocity shown in the intra-specific challenge.

**C** Entries to the starting point/intersection of the apparatus in response to each challenge presenting different stimulus fish. * on a solid line represents significant main effect of the species, * on bracket represent significant difference from the swimming velocity shown in the intra-specific challenge. **D** Percentage of time spent in the no/different stimulus (grey), the starting point/intersection (pink) or the stimulus zones (turquoise) in zebrafish (left) or paradise fish (right). * on brackets represents significant differences between time spent in the different zones. **E** Enter frequencies to the no/different stimulus (grey), the starting point/intersection (pink) or the stimulus zones (turquoise) in zebrafish (left) or paradise fish (right). * on brackets represents significant differences between time spent in the different zones.

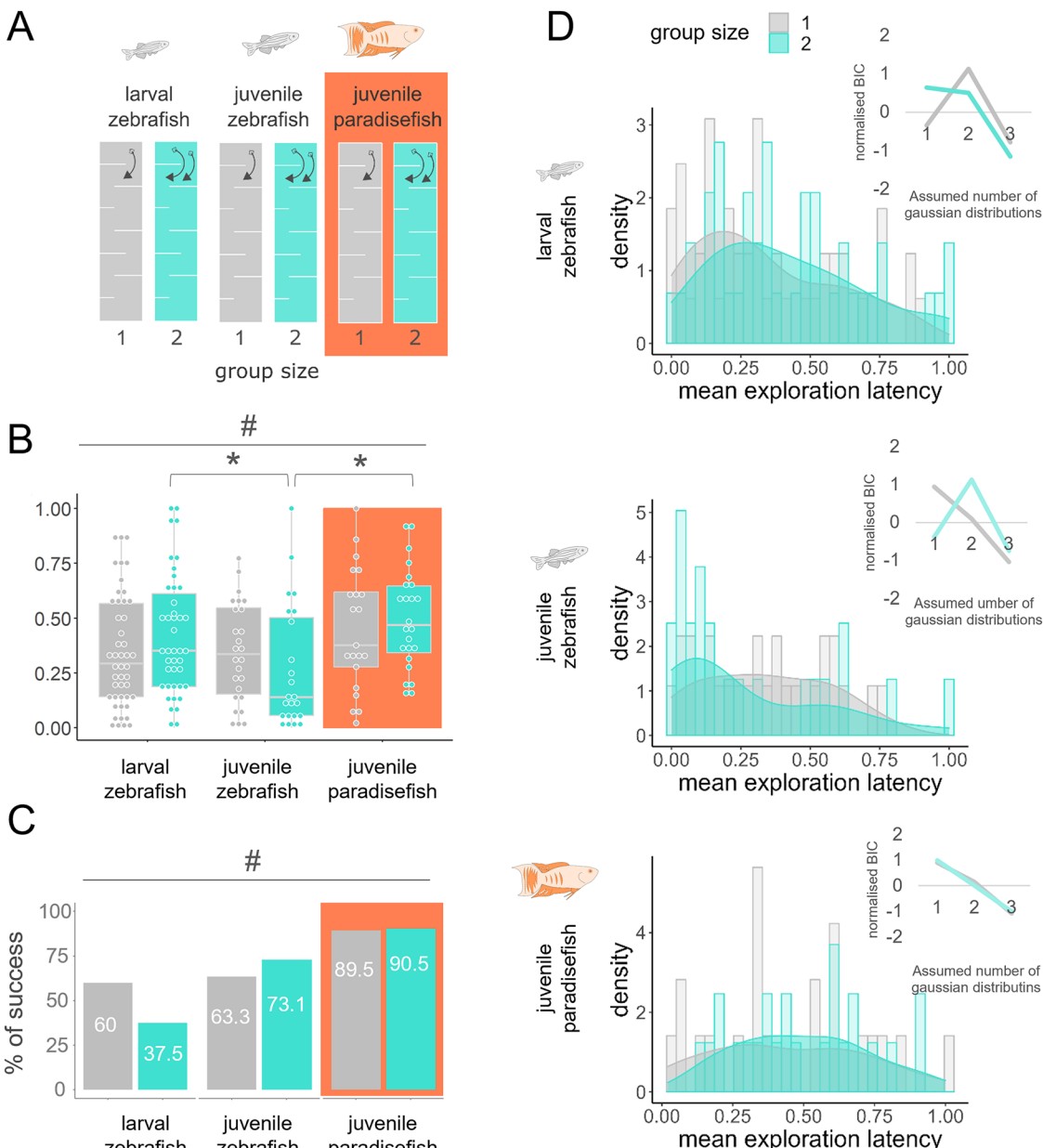

**Fig. 2 | Influence of conspecifics on exploratory behaviour. A** Experimental design of an exploration challenge using early (8 dpf) and late (30 dpf) larval zebrafish or paradise fish (species categories) applied alone or with a companion (group categories) to the slalom test. Colour code for the apparatuses matches the colour code of the plots. **B** Mean transition latency (scaled and averaged entry latencies to each chamber) of subjects. # represents significant main effect of the species category, *

represents significant difference between group categories. **C** Percentage of the animals that reached the last chamber in a 10 min long session. **D** Distribution of exploration latencies (histograms) and the probability of their uni-, bi-, or tri-modal nature. The highest BIC values show the most likely number of independent gaussian distributions that the total distribution may consist of.

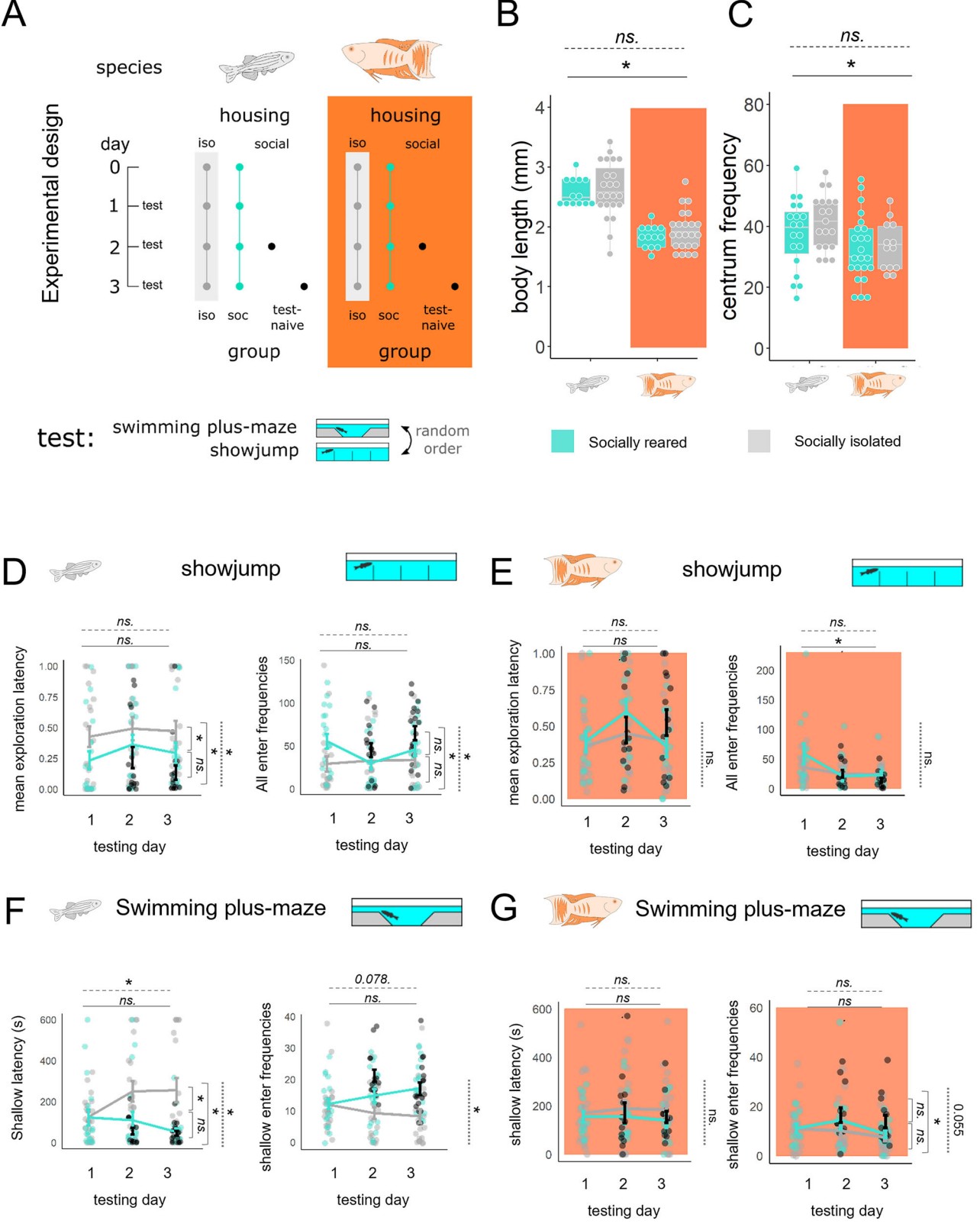

water surface, social isolation exerted slightly different effects on their endpoints. In contrast to SJ testing, behavioural changes in the SPM test were only present from the second day for the isolated individuals, indicating that only a relatively longer period of isolation was able to modify this exploration pattern. In paradise fish, subchronic social isolation had no

effect on anxiety-like behaviour in any of the tests. However, we noticed a significant decrease in exploration (frequency) in response to retesting in the species. In summary, explorative behaviour is highly affected by social isolation in zebrafish but not in paradise fish. All statistical parameters of the experiment are shown in Supplementary Table 3.

**Fig. 3 | The effect of social isolation on exploration. A** Experimental design: Zebrafish and paradise fish of late larval stage (27–30 dpf) were kept in isolation or in social groups for 4 consecutive days and their behaviour was monitored with the SPM and SJ tests. Additional test naïve controls were tested in each testing day to control for carry-over effects of repeated testing. **B** Body length of the two species in response to social isolation. * on solid line represents significant main effect of the species, * on dashed lines represents significant main effect of social isolation. **C** centrum enter frequencies in the SPM of the two species in response to social isolation. * on solid line represents significant main effect of the species, * on dashed

lines represents significant main effect of social isolation. Mean exploration latency and all enter frequencies of zebrafish (**D**) and paradise fish (**E**) in the showjump test. Shallow arm latency and shallow arm enter frequencies of zebrafish (**F**) and paradise fish (**G**) in the swimming plus-maze test. Horizontal solid or dashed lines with asterisk represent significant main effect of testing day or significant interaction between testing days and groups, respectively. Vertical dashed lines with asterisk represent significant main effect of the groups. Vertical braces with asterisks or different letters indicate significant post-hoc contrast between groups or testing days, respectively.

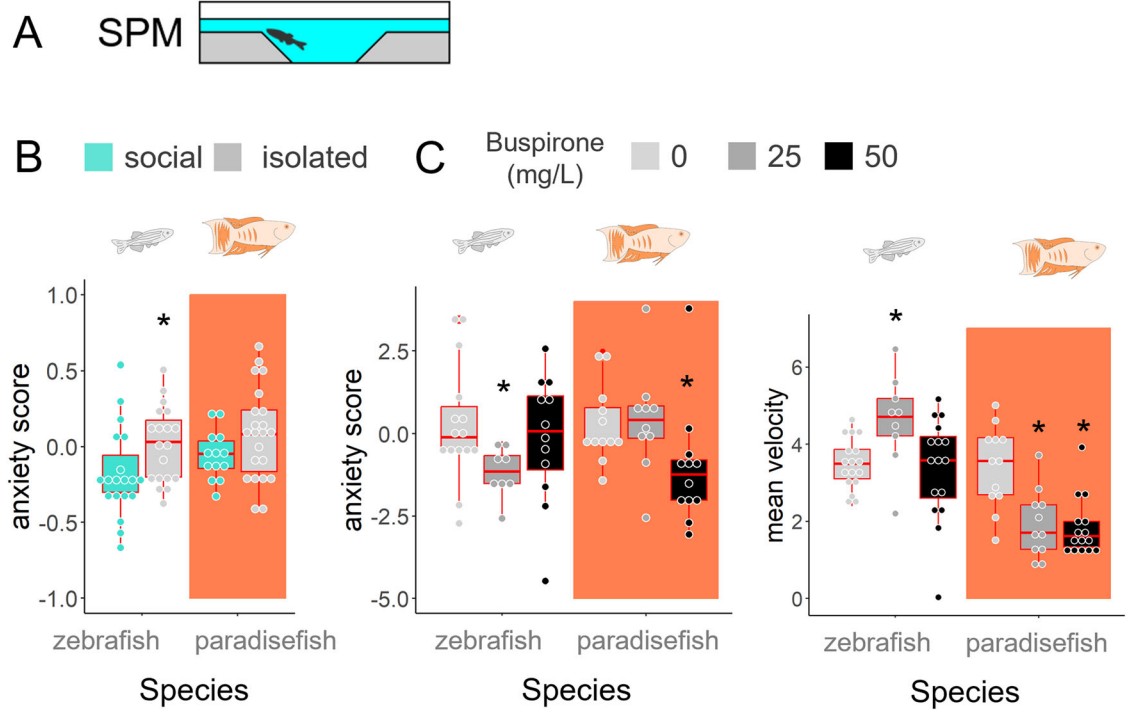

**Fig. 4 | Exploration variables as markers of anxiety. A** Zebrafish and paradise fish of late larval stage (30 dpf) were tested in the SPM test following social isolation stress or anxiolytic buspirone treatment. Anxiety scores were calculated from time and frequency data of shallow arm activity. **B** Shallow arm activity (anxiety-score) in response to 4 days of social isolation in zebrafish (white) and paradise fish (orange).

**C** Shallow arm activity (anxiety-score) and mean velocity in response to buspirone in zebrafish and paradise fish. Asterisks represent significant post-hoc contrast from socially reared (social isolation) or vehicle treated (buspirone) groups following significant group main effect.

## Experiment 4: paradise fish and zebrafish both express anxiety-like responses

Given that severe environmental perturbation (i.e. social isolation) did not influence the surface avoidance of paradise fish, we sought to assess whether this behaviour is associated with anxiety in this species at all. Therefore, we treated zebrafish and paradise fish of late larval stage (30 dpf) with the human anxiolytic medicine buspirone and measured their avoidance in the SPM test. We also show summary measures (averages of the three sampling days) of the SPM test from our prior social isolation experiment to compare the magnitude of change in anxiety-like responses to environmental and pharmacological perturbations. All source variables of the social isolation (Experiment 3) and the buspirone experiment (Experiment 4) are presented in Fig. 3 and Supplementary Fig. 1. We found that the same endpoint (anxiety score) that was not responsive to isolation in paradise fish (Fig. 4B), but in zebrafish decreased in response to buspirone in both species (Fig. 4C). Note that avoidance of zebrafish and paradise fish decreased in response to different concentrations of buspirone, 25 and 50 mg/L, respectively (Fig. 4C left). Interestingly, while zebrafish enhanced their locomotion in response to the anxiolytic agent, paradise fish decreased it (Fig. 4C right) indicating a similar dichotomy that was seen during a social challenge in our first experiment. These results indicate that surface avoidance behaviour is a marker of anxiety in both species, but in contrast to

zebrafish, in paradise fish it is not biased by acute social isolation. All statistical parameters of the experiment are shown in Supplementary Table 4.

## Experiment 5: paradise fish show increased behavioural consistency through time and contexts

In our next experiment, we compared behavioural consistency in zebrafish and paradise fish. Behavioural consistency implies predictable responses through time and context which we approached with intra-test (repeatability in time) and inter-test (repeatability in contexts) correlation analysis of behavioural endpoints, respectively. In the case of inter-test correlations we used both single time point (SiM) and refined summary measurements (averages of multiple sampling events, SuM) which better cover stable, trait-like features of the individuals[43]. Animals with more consistent behaviour show stronger correlations between test-retest performances (strong repeatability) or between different test performances (strong inter-test correlations) which can be further enhanced by refined sampling (e.g. using SuMs instead of SiMs). We sampled locomotion and anxiety-like avoidance behaviour in the SPM and open tank (OT) tests in three five-minute sessions with ten-minute inter-test-intervals (ITI) (Fig. 5A). We calculated between- and within-test variability (Fig. 5C), bootstrap analysis-based repeatability scores (Fig. 5D) and Spearman correlations (Fig. 5B) of several endpoints of the two tests. Swimming velocity and the number of immobile episodes

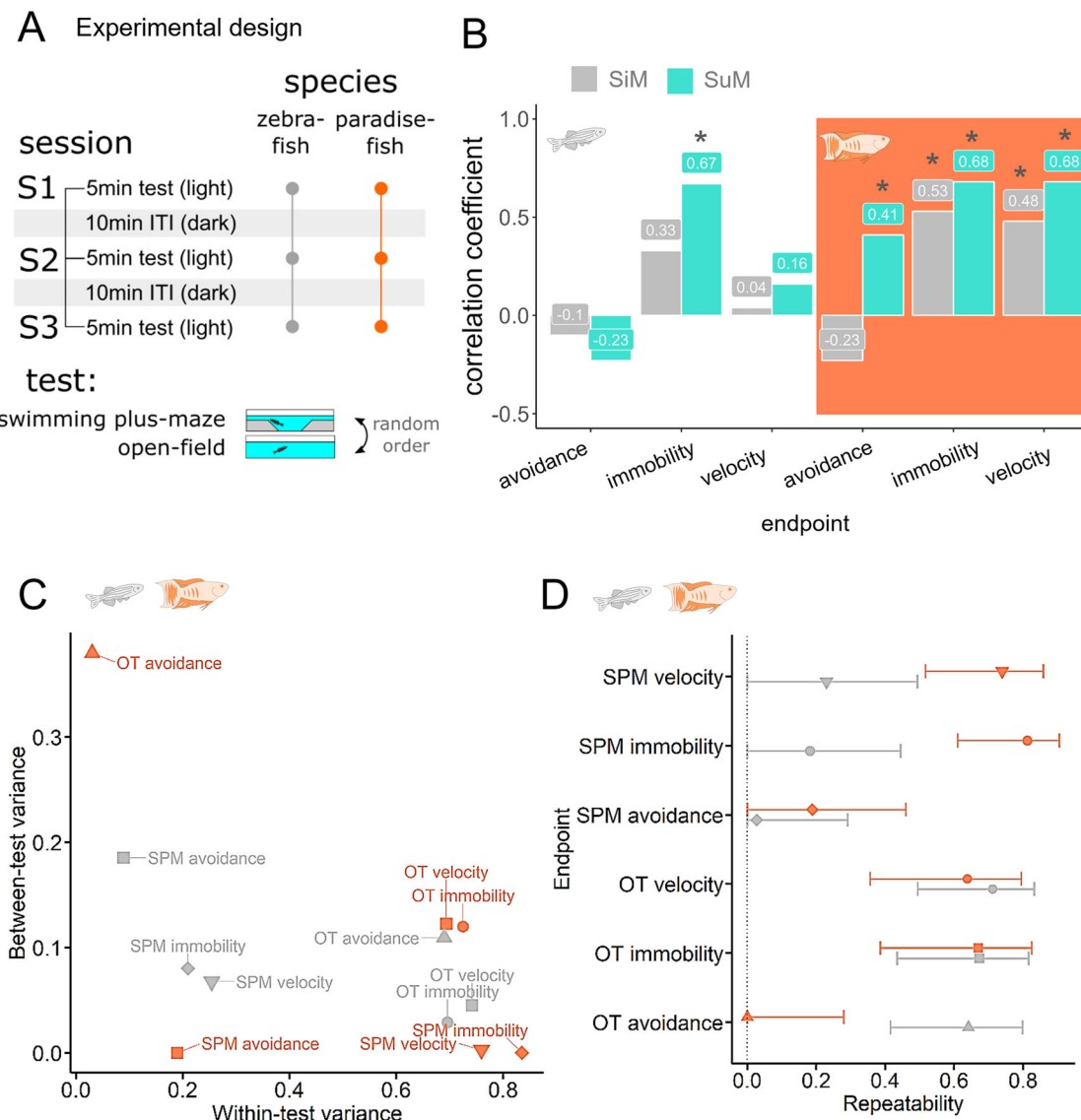

**Fig. 5 | Behavioural consistency in contexts and time were investigated by conducting two type of anxiety tests (SPM and OT) in a repeated design measuring inter and intra-test variability, respectively. A** Design of the experiment. **B** Inter-test correlations between similar variables measured in the SPM and OT tests. Grey bars represent correlations between single measures (one test event) and turquoise bars represent correlations between summary measures of three consecutive test events. **C** Single measures of the species in the between-test-within-test variance space. **D** Repeatability scores of single measures and their confidence intervals were calculated from between-and-within-test variability by parametric bootstrapping. Variables with a confidence interval crossing the 0 dotted line are not considered repeatable.

showed big within-test variability and small inter-test variability, consequently these readouts are the most repeatable in both species. In contrast, avoidance (from the shallow arms or the center of the SPM or OT tests, respectively) showed either low within-test variability or high between-test variability, making these variables less repeatable. In paradise fish, locomotor variables calculated either as SiMs or SuMs significantly correlated between the two tests, indicating a time-and-context-consistent locomotion (Fig. 5B). In addition, using summarisation, correlations between the two tests were revealed and further enhanced using avoidance and locomotor variables, respectively. Likewise, a significant correlation between the frequency of immobile episodes in the SPM and OT emerged by the summarisation of sampling events in zebrafish. In summary, paradise fish show consistent locomotor behaviour through time and contexts and consistent avoidance behaviour through contexts. In contrast, in the case of zebrafish, behavioural consistency is limited to the expression of episodic/saccadic swimming. All statistical parameters along with repeatability scores and confidence intervals of the experiment are shown in Supplementary Table 5.

## Experiment 6: paradise fish express different exploration strategy and level of working memory compared to zebrafish

Given these fundamental differences in exploration patterns, in our next experiment we aimed to assess by what strategy paradise fish and zebrafish *of late larval stage (30 dpf)* explore novel areas. We used the y-maze test, an aquatic version of the rodent working memory-assessment test under the same name[44,45] (Fig. 6A). The idea of the rodent y-maze is that intact working memory is necessary to apply the most effective strategy in exploration, namely the coherent alternation of different arms[44]. Both the rate of possible actions (indirect or direct revisits, alternations) (Fig. 6B) and the percent of alternations differed in the two species (Fig. 6C). Paradise fish show approximately 70% of alterations which exceed the number of alterations of zebrafish (*40%*) and is in the range of the performance observed for mice. Interestingly, higher alternation percentage is accompanied by lower swimming velocity in paradise fish compared to zebrafish. All statistical parameters of the experiment are shown in Supplementary Table 6.

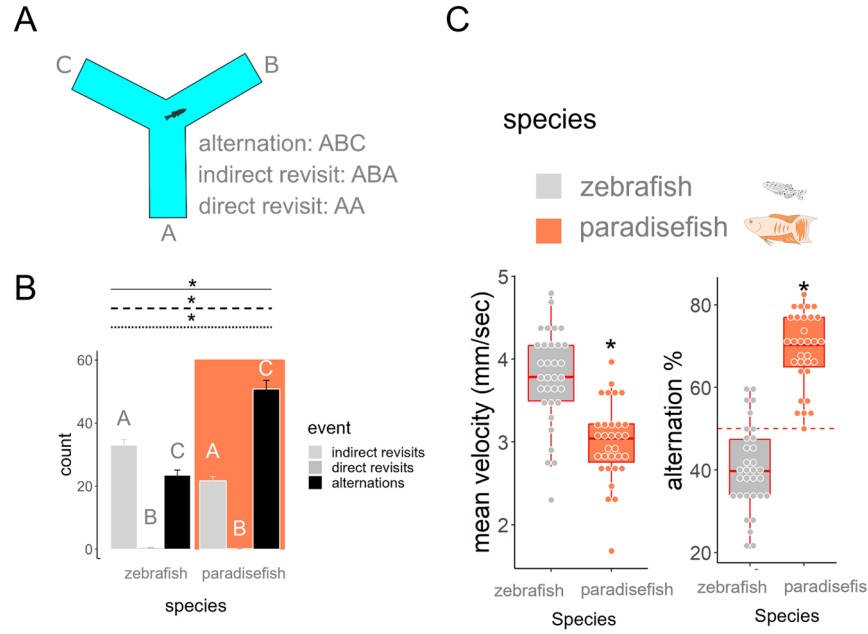

**Fig. 6 | Exploration strategy of zebrafish and paradise fish of late larval stage (30 dpf).**
**A** Schematic drawing of the y-maze platform and a list of the possible exploratory actions.
**B** Distribution of different exploratory actions in zebrafish (white) and paradise fish (orange). Asterisks on a solid, dashed or dotted line represent significant main effect of species, action type or significant interaction between these, respectively. Different letters represent significant post-hoc contrasts. **C** mean swimming velocity and alternation % of the two species. Asterisks represent significant species differences. Red dashed line represents 50% alterations, meaning random choice in the y-maze test.

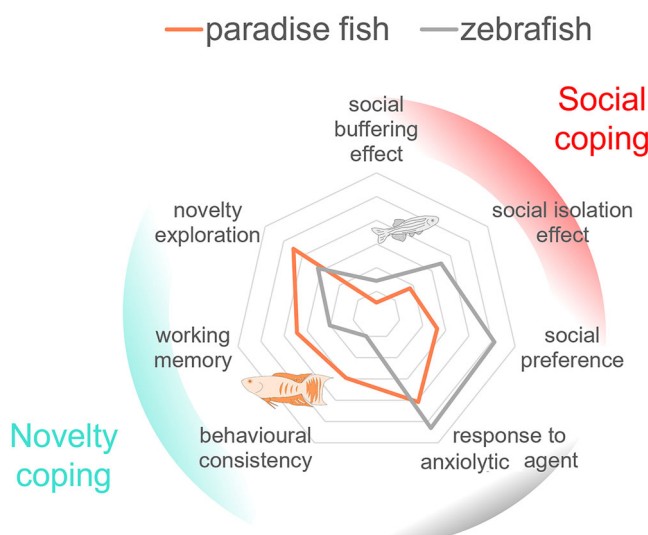

**Fig. 7 | Comparison of zebrafish and paradise fish of late larval stage in several assays tracing out the possible fields of use of the two species.** Effect sizes were plotted in the case of social buffering effect (mean latency), social isolation effect (average effect size of the shallow frequency and shallow latency of the SPM, and the total frequency, and the mean latency of the SJ test), social preference effect (time spent in social zone), response to anxiolytic agent (anxiety score), and working memory (alternation%). These were calculated by the subtraction of group means and the division by the total standard deviation of those. The absolute values of effect sizes were plotted. Averages of the inter-test correlation coefficients of summary measures were plotted in the case of behavioural consistency. Success rate in the slalom test were plotted in the case of novelty exploration.

## Discussion

Here we introduce the juvenile paradise fish model to improve individual-subject-based assays of exploration, anxiety and working memory. We show that the outcome of these tests is less biased by external factors when using paradise fish compared to zebrafish. We also show that this difference is likely due to zebrafish having a stronger social preference compared to paradise fish which can be an advantage in social paradigms but also a biasing factor when tested alone. The sensitivity of paradigms using the two species is summarized in Fig. 7.

## The impact of social stimuli on zebrafish and paradise fish

We observed that late larval-stage zebrafish show a strong baseline preference for conspecifics—a robust and well-documented effect across multiple age groups, test paradigms, and studies[18,19,23,46–49]. In contrast, our findings show for the first time that this preference persists, albeit less prominently, under perturbed conditions—specifically, when a paradise fish is also presented as a stimulus. The presence of a paradise fish alone did not elicit a strong approach response from zebrafish, suggesting that their social attraction behaviour is conspecific-specific. This is consistent with previous work by Engeszer and co-workers, which demonstrated that zebrafish develop specific preferences, even based on pigmentation patterns, early in development[47].

Paradise fish, in contrast, did not display a consistent preference for conspecifics under any of the tested conditions in Experiment 1, and often preferred the empty zone when it was available. However, during the double-species challenge, they spent more time near the zebrafish individual. This behaviour may reflect either avoidance of the conspecific or a preference for the heterospecific. Avoidance seems unlikely, as paradise fish did not avoid conspecifics in the conspecific-only condition, and prior studies suggest conspecifics can serve as social reinforcers in this species[33]. A more plausible interpretation is a conditional preference for the zebrafish, possibly elicited by curiosity—modulated by interspecific-induced anxiety and conspecific-induced social buffering.

Notably, in Experiment 1, paradise fish increased their swimming speed in response to the presence of a zebrafish, whereas in Experiment 4, their swimming speed decreased following administration of the anxiolytic agent buspirone. This pattern suggests that elevated swimming velocity may serve as a behavioural marker of anxiety-like states in paradise fish, and that the presence of a zebrafish individual may induce such a state. However, this anxiety may be mitigated by the presence of a conspecific, indicating a potential social buffering effect.

Social buffering of anti-predator behaviour has previously been observed in late larval-stage paradise fish, as well as across diverse fish species in response to stressors ranging from novelty to predation cues[22,38,39,41]. These effects have been documented in solitary individuals[50], loosely aggregating species[21,22,51], and those with defined social structures[41,52], suggesting that social buffering is a conserved mechanism with a general stress-ameliorating function across fish taxa.

We also observed evidence of social buffering in late larval-stage zebrafish, which initiated exploration of the slalom test more rapidly in the

presence of a conspecific, compared to age-matched paradise fish or younger zebrafish larvae. While previous studies have linked social buffering in zebrafish to reduced fear-related behaviours[22], our findings are the first to demonstrate its influence on exploratory behaviour and anxiety-related responses.

Interestingly, while paradise fish did not exhibit a bias in response to the presence of a conspecific during the slalom test, both early-stage and late-stage larval zebrafish demonstrated bimodal patterns of exploration when tested alone and in pairs, respectively. This suggests the existence of multiple motivational states and/or exploratory strategies within the population of 8 days post-fertilization (dpf) zebrafish when tested individually, and within the population of 30 dpf zebrafish when tested in dyads. We speculate that 30 dpf zebrafish with a strong social preference may exhibit a pronounced, unimodal exploratory pattern when tested alone, as they are highly motivated to locate conspecifics. In contrast, when tested in pairs, the exploratory behaviour of these older larvae may be influenced by the nature of their interaction – whether affiliative or antagonistic – with the conspecific present. For early larval zebrafish, the observed bimodality might reflect heterogeneity in social development: some individuals may already exhibit a preference to seek out conspecifics, while others remain more socially plastic and engage in solitary exploration. Heterogeneity in social development at early stages were previously reported in zebrafish[46,53].

Zebrafish responded to short-term social isolation with increased latency to explore and reduced exploration of aversive zones in both the SPM and SJ anxiety tests. Notably, both the number of centre entries in the SPM and body size were unaffected, suggesting that the observed changes reflect anxiety-like behaviour rather than a general reduction in exploratory activity or physical condition.

Previous studies have reported that social isolation beginning at early stages reduces locomotion and exploration at the late larval stage[19,26,46], and we also showed before that isolation from 14 dpf can delay exploration, reduce stress-induced arousal, and increase anxiety-like behaviour in 30 dpf zebrafish[23]. However, our current study is the first to demonstrate similar effects following a much shorter isolation period. It is also the first to describe the developmental emergence of these effects.

In contrast, paradise fish did not exhibit behavioural changes following social isolation, an effect demonstrated here for the first time. However, they showed decreased exploration upon repeated testing, which is likely due to a carry-over effect of prior experience - similar to the phenomenon of one-trial tolerance described in rodents[54,55], tough not yet confirmed in fish[56]. It is important to note that our buspirone experiment indicate that paradise fish do display anxiety-like responses; however, these responses were not exacerbated by social isolation.

### Exploration strategies in zebrafish and paradise fish

Despite their slower swimming speed and smaller body size at the tested developmental stage, a higher proportion of paradise fish completed the Slalom test compared to zebrafish (Fig. 2C). To find out how robust this seemingly counterintuitive trend is, we directly compared the two species locomotion, general exploration, and stimulus exploration across social, anxiety-related, and cognitive tasks (Supplementary Fig. 2, Supplementary Table 7). We found that slower swimming is accompanied by higher exploration in most cases. This indicates that differences in body size and locomotion does not limit paradise fish in exploration, hence are comparable to zebrafish, and that they might use a different strategy to explore novel areas. A previous study also suggested that paradise fish show a more efficient strategy that enables the exploration of significantly more novel compartments than simulated random walks[57].

In the behaviour consistency experiment (5) behavioural variables of zebrafish were poorly correlated between different test types and between different sampling occasions of the same test type, indicating that their responses to novelty are highly affected by the type and timing of the testing[43]. This is also supported by the low repeatability scores of these variables. The only significant correlation between test types was the

number of immobile episodes, which possibly reflects that all novel conditions can trigger some extent of freezing behaviour in zebrafish. In contrast, paradise fish showed significant between-test correlations in immobility, swimming velocity and avoidance too, and more significantly repeatable variables, meaning that these traits are expressed consistently and are less biased by the timing and type of the test. Consistently expressed behaviour is also indicative of a solid strategy for exploration in the species.

In the classical rodent Y-maze, working memory is typically assessed by the percentage of alternations, ranging from 50% (random choice) to 100% (perfect alternation)[44]. In our study, paradise fish and zebrafish showed approximately 70% and 40% alternations, respectively. This suggests that paradise fish exhibit functional working memory, whereas zebrafish performance falls below the threshold of random choice, indicating distinct exploratory strategies between the species. Forty percent alternation is also supported by the free-movement pattern y-maze test[45] usually applied in zebrafish. Analysis of all possible action patterns revealed that neither species explores randomly. Paradise fish behaviour was dominated by true alternations (ABC), while zebrafish primarily showed indirect revisits (ABA). These patterns may reflect a novelty-seeking strategy in paradise fish versus a home-base-oriented strategy in zebrafish. Given these differences, direct comparisons of working memory between the species may not be appropriate. However, the strategy employed by paradise fish appears to demand greater working memory engagement.

### Summary

In summary we found that zebrafish has a strong and specific social preference and while the presence of a conspecific alleviates anxiety-like behaviour, the absence of it enhances it. In contrast, paradise fish were not affected by any of these conditions. Apparently, the social behavioural repertoires of paradise fish and zebrafish show only slight overlaps, furthermore, the expression of these behaviours are more conditional in the former species compared to the latter one. The exploratory behaviour of paradise fish and zebrafish in individual settings is also different: paradise fish express a novelty seeker strategy that is consistent in time and context and requires working memory, while zebrafish responses are more effected by external conditions when tested alone. These differences are likely to stem from the differences in the social structure of the two species: while zebrafish only live in shoals, paradise fish exists in looser aggregations. These differences in the natural history and associated biological traits of the two species help us to understand the most effective use of these species in preclinical settings (Fig. 7).

### Limitations of the study, future perspectives

We acknowledge a key limitation of this study: paradise fish have traditionally served as a model in ethology rather than translational neuroscience, whereas most of the tests and paradigms applied here were originally developed for zebrafish. As a result, relevant prior literature is limited, and the robustness and generalizability of some findings remain uncertain. Nonetheless, we hope that this study – alongside the recent publication of the paradise fish reference genome[27] and advances in husbandry practices[28] – will support broader adoption of this species in neuroscience research. Given the behavioural and neurobiological differences between zebrafish and paradise fish, we propose that paradise fish could serve as a valuable complementary model in translational studies, potentially enabling clearer interpretation of certain outcomes, especially in relation to problem-solving behaviour involving learning and cognition.

### Materials and methods
#### Animals
Wild-type zebrafish (AB) and paradise fish lines were maintained in the animal facility of ELTE Eötvös Loránd University according to standard protocols[28,58]. Experimental subjects were male and female animals aged between 10 and 30 days post fertilization (dpf). Adult paradise fish were maintained as described before (PMID: 33799915), in glass tanks with weekly water changes. The conditions were: pH 6.8–7.5, conductance:

350–450 μS (the slight increase is due to evaporation), temperature: 25–27 °C. Nitrite, nitrate and ammonia concentrations were measured regularly, and they were within the following limits: nitrite 0.05–0.08 mg/L, nitrate 10-15 mg/L, ammonia: less 0.05 mg/L. All fish were maintained in a standard 14 h/10 h light/dark cycle. Feeding of zebrafish and paradise fish larvae started at 5 dpf with commercially available dry food (a 1:1 combination of <100 μm and 100–200 μm Zebrafeed, Sparos). After 15 dpf, juvenile fish were fed using dry food with gradually increasing particle size (200–400 μm Zebrafeed, Sparos) combined with fresh brine shrimp hatched in the facility. Fish larvae of both species were moved from the incubator to the facility at the age of 5 dpf, where they were kept in 2.5 L containers. At the beginning stocking density is 40 fish/tank, at one month post fertilization we reduced the density to 20–25 fish/tank, and around two months of age to 10–12/tank. Fish were all socially reared except for one experiment (social isolation experiment) where half of the subjects were subjected to isolation for 3 days. Animals were anaesthetised with lidocaine and terminated with tricaine overdose (400 mg/L) immediately after each experiment, as required by Government Decree no. 40/2013. (II. 14.). All protocols used in our study were approved by the Hungarian National Food Chain Safety Office (Permits #PEI/001/1458-10/2015, #PE/EA/2483-6/2016 and #PE/EA/406-7/2020). We have complied with all relevant ethical regulations for animal use.

### Environmental modifications

Sub-chronic social isolation of subjects was conducted between 27 and 30 dpf. Animals originating from the same spawning were randomly allocated to social rearing or social isolation for both species. Isolated animals were kept in white opaque plastic tanks (52 × 35 × 46 mm, depth × width × length) depriving the individuals of sensory cues from conspecifics[23]. Tanks were filled with fish system water, half of the volume of which was replaced on a daily basis. Socially reared control animals were subjected to similar conditions, with the exception of the size of their aquarium, which was matched to the greater number of larvae, providing ~50 ml volume for each individual.

### Drug treatments

Buspirone (Sigma Aldrich, Cat. no. 33386-08-2) was dissolved in system water and administered as a water bath for 10 min followed by a 5 min washout in compartments of a 12-well plate followed by a behavioural test immediately. Each compartment contained 1.5 ml of treatment solution. We applied 0 (vehicle), 25 and 50 mg/L concentrations, which were selected based on our previous studies and pilot experiments[23,59].

### Behavioural procedures

Social exploration tasks. To measure exploration in social context the sociability test of the Dreosti laboratory[18] and the slalom test developed by our group (see below) were used.

Sociability test. The social preference test apparatus is a U-shaped platform (40 × 32 mm) consisting of two identical arms, made of white opaque material (VeroWhite PolyJet Resin), with glass window partitions enabling only visual communication between the fish (Fig. 1A). The test is based on the visually-guided preference of zebrafish toward conspecifics. A fifteen minute habituation period followed by a fifteen minute challenge period, without or with the presence of a stimulus animal, respectively, according to the protocol from the original paper and confirmed by our previous studies[23]. Both focal and stimulus animals were naïve to test. Time spent in the social (stimulus conspecific) and non-social (no/or different species) zone and enter frequencies to the intersection (starting point) were measured.

Slalom test. The slalom test was recently developed by our laboratory to measure the the exploratory drive of larval or juvenile fish, particularly in a social context. The relatively long test apparatus (15 × 20 × 85 mm, depth × width × length) consists of 12 equal-sized chambers, each visible only from the adjacent ones (Fig. 2A). This way the test offers enough novelty for the fish and separates the population into individuals who did and individuals who did not succeed to reach the last chamber. One or two individuals per maze were placed in the first chamber then their behaviour was recorded for ten minutes. Enter latencies (sec) to each chamber (2nd, 3rd, …12th) were scaled for the investigated population from 0 to 1 then these were averaged between chambers resulting in the variable "mean transition latency" (0–1). Percent of success means the proportion of the group that successfully reached the 12th chamber. Sessions were analysed up to 10 min, or until the (last) individual reached the last chamber.

Novelty exploration tasks. To measure avoidance responses to novelty the swimming plus-maze (SPM), the showjump (SJ) and the open tank (OT) tests were used.

Swimming plus-maze test. The SPM test from our laboratory Varga et al.[59] is a +-shaped platform consisting of two shallower (8 × 10 × 2.5 mm) and two deeper arms (8 × 10 × 5 mm), different in depth, connected by a center zone (8 × 8 × 5 mm) (Fig. 3A). Larval and juvenile zebrafish prefer the deep over the shallow arms and the center zone, a preference linked to anxiety-like motivational states. Ten-minute-long (experiment 3 and 4) and three five-minute-long (experiment 5) tests were conducted. Time spent in every zone, the latency to enter each zone and the mean of overall velocity were measured. Standard anxiety scores were calculated as scaled time minus scaled latency multiplied with -1, representing a positively correlated measure of anxiety-like behaviour in these tests. For the validation procedure and detailed specification of the test, see Varga et al.[59].

Show jump test. The SJ test (15 × 20 × 42 mm, depth × width × length) consists of 4 equal-sized chambers separated by 10 mm high walls (Fig. 3A). Subjects need to swim above the wall, close to the water surface to enter a novel chamber. The behaviour in the test, similarly to the behaviour in the SPM, is based on the conflict between explorative motivation and novelty-induced anxiety-like state. Ten-minute-long tests were conducted. The number of all chamber entries and mean trasition latency were measured similarly as in the case of the slalom test.

Open tank test. OT tests were conducted in 6-well-plates ($d = 35$ mm). The behaviour in the test is based on the natural aversion of highly exposed, open areas. An inner circle that covers eighty percent of the total volume was traced out representing the aversive central area. Proportion of time spent in the periphery, the number of immobile episodes, and overall mean velocity were measured.

Y-maze test. To characterise exploration strategies and measure working memory the y-maze test was used. The aquatic y-maze consists of three equal-sized arms (each 5 × 7.5 × 30 mm, depth × width × length) (Fig. 6A). Ten-minute-long tests were conducted. We recorded alternations (three consecutive visits are done to different arms), direct revisits (two consecutive visits are done to the same arm) and indirect revisits (following two consecutive visits to different arms the third is done to the first arm). The sequences of these events were calculated in a sliding window, meaning that following the third visit every subsequent visit defined a new sequence.

### Experimental design and analysis

In *Experiment 1* we aimed to compare the sociability of the paradise fish and zebrafish in different conditions. We observed either 30 dpf paradise fish ($n = 15, 17, 16$) or zebrafish ($n = 14, 14, 15$) as they faced a conspecific (intraspecies challenge), an individual of the other species (interspecific challenge) or individuals of both species (double challenge) in the U-shape sociability test (Fig. 1A). Sample sizes, and the different challenges what those are assigned to are mentioned in the same order in both species. Six

fish were tested simultaneously in separate apparati. The intraspecific challenge is an indicator of the social activity of each species in baseline conditions while the double challenge measures the same during a mild perturbation, i.e., the presence of the other (non-predatory) species. Intraspecific challenges are controls to determine whether changes in the other conditions are species-specific e.g., triggered by a conspecific. Stimulus animals were naïve to the test and were previously housed in different tanks. The behaviour of the subjects was video recorded.

In *Experiment 2* we aimed to compare the exploratory drive in social and non-social contexts in paradise fish and zebrafish. We applied either 30 dpf paradise fish ($n = 21$, 19, for social an non-social contexts, respectively) or 8 dpf ($n = 24$, 30) or 30 dpf zebrafish ($n = 26$, 30) to the slalom test individually or accompanied by a conspecific (Fig. 2A). The behaviour was recorded for 10 minutes.

In *Experiment 3* we aimed to compare the effect of sub-chronic social isolation on anxiety-like states of paradise fish and zebrafish. We allocated paradise fish ($n = 13$, 23) or zebrafish ($n = 19$, 19) to social or isolated housing, respectively, at day 0 (27 dpf), then assessed their behaviour from day 1 to 3 in the SPM and SJ tests for 10 minutes each day. To control for a carry-over effect of repeated testing we applied additional test naive paradise fish ($n = 13$, 14) and zebrafish ($n = 15$, 18) individuals from day 2 to 3 (Fig. 3A). Sample sizes, and the different conditions what those are assigned to are mentioned in the same order in both species. The behaviour was video recorded.

In *Experiment 4* we aimed to validate whether surface avoidance behaviour can be considered as a marker of anxiety-like states in paradise fish. We treated either 30 dpf paradise fish ($n = 12, 12, 15$) or 30 dpf zebrafish ($n = 17, 9, 16$) with 0, 25 or 50 mg/L concentrations of the clinical anxiolytic buspirone and measured their surface avoidance behaviour in the SPM. Sample sizes, and the different concentrations what those are assigned to are mentioned in the same order in both species.

In *Experiment 5* we aimed to compare behavioural consistency through time and contexts of 30 dpf paradise fish and 30 dpf zebrafish. We used either paradise fish ($n = 24$) or zebrafish ($n = 24$) to three sessions of OT then three sessions of SPM for five minutes each, interrupted by 10 min inter-session-intervals (Fig. 5A). OT and SPM testing sessions separated by 45 min inter-test-intervals when fish were kept in their home-tank.

In *Experiment 6* we aimed to assess the exploration strategies and compare the working memory of paradise fish and zebrafish. We placed either 30 dpf paradise fish ($n = 31$) or zebrafish ($n = 32$) in the aquatic version of a y-maze test for 10 min.

The behaviour of the subjects were recorded in a Zantiks MWP unit (Zantiks Ltd., Cambridge, UK) in $640 \times 480$ resolution in 30 fps and analysed with Noldus Ethovision XT[60] in each experiment.

## Reporting summary
Further information on research design is available in the Nature Portfolio Reporting Summary linked to this article.

## Statistics and reproducibility
Statistical analysis was done using the R statistical environment[61]. Hypothesis testing on multiple group design experiments was done using linear models (ANOVA or Kruskal-Wallis rank sum test) followed by post-hoc contrasts (*t*-test or Wilcoxon test) depending on the distribution of data, adjusted with false discovery rate approach[62]. Hypothesis testing on single treatment group design experiments was done using Student's *t* tests. Sample sizes (see "Experimental design and analysis" subsection) indicate biological replicates.

Correlations were calculated using the Pearson method with the *rcorr* package. Repeatability analysis was done according to the analysis pipeline of Answer et al.[63] for each test-type by calculating the proportion of within-test variance out of total variance, as shown below: within test variance/ (within test + between test variance), where within-test (or between-individual) variance means the variation of the group's behaviour within a given test, and between-test (or within-individual) variance means variation of

individuals' behaviour across weeks (test repetitions) for a given test-type. The R package *rptR* (version 0.9.22) was used to calculate variances and repeatability estimates similarly to previous investigations. The function calculates 95% confidence intervals of estimates by parametric boot-strapping, with the number of parametric bootstraps for interval estimation set to 1000. Estimates with confidence intervals that did not include 0 were considered statistically significant. After examining normality of variables, the variances, adjusted repeatability estimates and their uncertainty were calculated for all behavioural variables (avoidance, immobility, velocity). Gaussian mixture modelling was done using the R package *mclust* on the mean transtition latency variables. The modality of distribution was decided based on the highest BIC value. BIC values were scaled to 0–1 in each group and ploted together to show the differences in distribution types between species of particular age groups.

Detailed statistical analysis is included in the Supplementary Table.

## Data availability
Behavioural datasets that support the results presented in the paper are available on Zenodo: https://zenodo.org/records/14028710[64].

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

## Acknowledgements
We would like to thank Anita Rácz for fish care. This work was supported by the ELTE Eötvös Loránd University Institutional Excellence Program Grant 1783-3/2018/FEKUTSRAT. E.M. is funded by Project no. RRF-2.3.1-21-2022-00011, titled National Laboratory of Translational Neuroscience which has been implemented with the support provided by the Recovery and Resilience Facility of the European Union within the framework of Programme Széchenyi Plan Plus. M.V. is a János Bolyai fellow of the Hungarian Academy of Sciences (BO/00555/22/8).

## Author contributions
Conceptualization: Z.K.V., D.P., E.M., A.M. and M.V. Funding acquisition: M.V. and A.M. Investigation: Z.K.V., D.P. and T.C. Methodology: Z.K.V. Writing – original and revised text: Z.K.V., E.M., A.M. and M.V.

## Funding

## Competing interests
The authors declare no competing interests.
