## [Transparent Peer Review file · Communications Biology]

Paradise fish (*Macropodus opercularis*) as a complementary translational model for emotional and cognitive function

Corresponding Author: Dr Máté Varga

Version 0:

Reviewer comments:

Reviewer #1

(Remarks to the Author)

This paper by Varga et al was a stimulating read. The authors report interesting and novel behavioural data highlighting differences between young zebrafish and paradise fish to establish and further demonstrate utility of paradise fish in research. They considered three main behavioural contexts: exploration, anxiety, and working memory, that may provide potential of modeling for various neuropsychiatric disorders and basic neuroscience research using paradise fish. Species-specific differences in social interaction, non-social exploration, aversion/desirability to novelty, and responses to anxiety and pharmaceutical anxiolytic agents are important scientific investigations, for both translational neuroscience researchers and broader academic community. Using individuals and paired fish for same tasks and designing novel tasks [e.g., Slalom test] is exciting and commendable. The focus on two developmental stages is also a strength of the manuscript though that comes with caveats of limited ability to make inference about any effect of sex and about the mature forms of behaviours in the two species. I enjoyed reading the manuscript and applaud the authors for their good work. I have some more serious queries and comments, along with minor and easily fixable suggestions emphasised below. Please considering these clarifications and changes as I believe these can enhance confidence in the study's findings.

General Issues

As is the case in such studies, all of the behaviours/variables reported are movement-based, which can be affected by the relative size of the fish, for both within and inter-species comparisons. There can be large variation between siblings raised together and isolated fish can be larger within the first 3 weeks of life as they do not face much competition for food whereas social rearing leads to some monopolizing of food by some siblings. It would be really helpful to have some indication [mean and/or range of body length] of the size of both species as the same arenas are used for all fish of the two species and the same dimensions may represent different level of difficulty [e.g. arena size of 20X body length may induce different level of anxiety than 15X]. Potentially some of the variation could be due to size differences between zebrafish and paradise fish [and social vs. isolated fish of same species; and within same treatment siblings as well]. Please include a figure or table for size of fish; it would help with interpretation of results.

The reporting of findings and later discussion would also benefit from reduction in generalization and toning down some of the less supported claims. While the authors focus on the dichotomy between the two species, in parts of the manuscript and in similar literature, fish [and other animals] can respond to anxiogenic/anxiolytic factors by increasing OR decreasing locomotion and even both in different ways within a short time [e.g. freezing with bouts of fast/erratic movements]. A more comprehensive understanding would be possible for the reader if locomotion is not defined as an axis but rather a grid or plane where the same treatment can affect both hyperactivity and freezing. To this end, while I appreciate the authors providing composite score for anxiety, exploration, etc., please consider providing simpler movement [distance traveled/velocity, or distance from the anxiogenic stimulus/side] over the whole time of testing as supplementary information in this and future manuscripts to aid with interpretation of the results. I think this would make the findings more interesting and reinforced.

Abstract

L24: Given lack of aggression testing in the manuscript, I would encourage authors to rephrase 'defensive behaviours' to a more precise term regarding risk-avoidance in novel environment or protection from predator in their behavioural testing.

Introduction

L49: The authors use the phrase 'such a simple social structure' to define natural shoaling tendency of zebrafish. I find that

difficult to follow as shoaling is anything but simple. With competition for resources, dominance hierarchy, disease, and other psychological and health dynamics, etc., some are mentioned by the authors in the next sentences, shoaling can be quite complicated. Could they elaborate on this and/or provide a more suitable phrase. Also, it would be useful to know if their 'simple' designation of zebrafish shoaling is in comparison to solitary lifestyle or other more/less interactive social structures.

L63: Again, please use phrasing that is more accurately representative of antipredator behaviour. 'Defensive' the broader term used here, implies both conspecific aggression/competition and protection from predators and other harm.

L65: Please provide more support here for 'one that resembles more human social settings'. Do the authors imply this only for more individualistic societies? What specific social behaviours or dynamics make paradise fish more similar to human socialization. As one could also argue the opposite that more social species like zebrafish bear more similarity to humans, particularly collectivistic cultures so such an elaboration here [1-2 sentences] would be interesting and provide a more comprehensive understanding for the readers.

L66: What does function imply here as various important life necessities are already listed earlier as social activities. Please list elements of life that are performed individually by paradise fish. It would also be helpful to know if paradise fish show any parental behaviour [nest building and guarding embryos and young] and for how long. Can they be classified as precocial or altricial? When do young paradise fish become solitary? Brief mentions of these behaviours would help with better comprehension of the mentioned 'challenges' in the next sentence as well.

L76: Could you describe here what is meant by 'human phenomena'?

Results

L86-89: This description could be more helpful if it mimics the same order as the Figure 1A-C [Cons, I-Spc, Both].

L90: Figure 1B would allow better comparison if the y-axis is the same for the two graphs. Is the obvious size difference in the images of zebrafish and paradise fish on the top of Figure 1B and 1C [and later figures as well] intentional and represented of tested fish? As mentioned in the comment above, size of the fish can affect many of the tested behaviour in the same arenas, please elaborate if there was a difference between the two species in size at pfd8 and pfd21. Please also comment on the variability in size within each species at these ages.

L90-Figure 1C: It is difficult to draw conclusions about sociability from the '% time in chamber' variable alone. It is common to use the time values for the two zones in this task and calculate before/after social preference index [SPI] as Dreosti et al [2015; <https://doi.org/10.3389/fncir.2015.00039>] did and others since with this type of protocol and data. I am also concerned that some of the fish spent 0 or 100% of time in one or the other zone. Were these fish frozen/immobile in a zone or did they explore the whole testing area at some point to have awareness of other chambers? These graphs are difficult to decipher without the comparison of habituation vs post-social stimuli and if the exclusion criteria don't include exploration of the whole testing area. Please include more details on this in the manuscript.

L90-99: Please indicate the age of the fish on the figure and the caption. I recommend being consistent with terms for the arena parts. It is not clear if 'zone' is the same as 'chamber' or if they represent different parts. Figure 1B is difficult to interpret as the left panels lacks any indication [* or ns] shown in 1C. Is the velocity for the whole 30-minutes or just the 15-min habituation, or the 15-min of the social test phase?

L96-97: It may be helpful to state that * represents significant difference from Conspecific condition [C-spc]. The current phrasing makes the use of * difficult to follow here.

L97-99: In the figure caption, please indicate if the values in Figure 1C represent time during the second half of the test or the whole test? Please consider using 'social zone' instead of chamber in figure 1C.

L100-103: The graphs in Figure 1B show higher values and variation for zebrafish than paradise fish, which is not given much consideration at all in the manuscript until a brief mention in the discussion. Please state expect range of swimming speeds for the two species and include if young zebrafish have been reported to swim faster than paradise fish in literature or if this is just a unique finding or artifact in your study.

L111-116: The interpretation portions of this summary [lost preference, developed aversion, etc.] would be better suitable in the discussion. Please only summarize the data trends here.

L118-119: This sentence needs citations. Please cite existing studies that first established and have since demonstrated social buffering in zebrafish and other social species.

L123: Please indicate what age is meant by larval zebrafish and juvenile here.

L128-131: The interpretation parts of this sentence should be avoided in the results section and be placed in the discussion.

L134: BIC should be written out as bayesian information criterion and briefly defined at first use.

L139: Figure 2D would allow better comparison if the y-axis is the same for the graphs. In general, lack of data from pfd8/larval paradise fish makes the pfd8 zebrafish data less useful, and the figure more distracting and complicated. Larval zebrafish are also much smaller at pfd8 [compared to pfd21] and the same testing arena may have presented a much greater challenge for pfd8 zebrafish than pfd21 zf and pf. I'd encourage the authors to take out pfd8/larval zf data from this figure and analysis, and focus only on the pfd21 zf and pf comparison.

L142: Please provide some motivation here for why this specific amount/level of isolation [3 days] was chosen; as compared to more commonly used 1 hour, 24 hours, 7/14/21 days that have been previously used by the authors [Ref 24] and many others.

L153-160: These results are interesting as these are in contrast of published reports of higher exploration and activity in open-field type testing following isolation of young zebrafish. Did the authors find any difference in overall locomotion [not exploration but mobility level] of fish after isolation?

L166-169: Figure 3 is difficult to see due to the size of font [even if one can ignore the line numbers superimposed on top of the right side of the figure where significance is indicated with vertical bars]. Please consider changing the size and orientation of this figure for better visibility and briefly summarizing the significant findings, trends, and general direction of the results in the caption.

L182: Consistent with previous two sub-sections in the results, this sub-section would benefit from a summary sentence at the end of this analysis.

L193-202: These results are interesting and I'd encourage the authors to keep the interpretation elements for the discussion.

Generally, the reporting of findings and later discussion can also benefit from less generalization. L203-Figure 4B and 4C. The colour combinations and y-axes are confusing here. Please use the same range for y-axis for B and C. Consider using different combinations of colours to indicate isolation treatment [e.g., grey & teal used in Figure 3] and buspirone treatment. It is not clear just by looking at the figure 4C which fish were isolated and which were not. L221: Where were fish placed during the ITI? Home-tanks or new tanks? L236: Figure 5C is difficult to see. Please make the grey bars and labels darker and more visible. L252-255: I'd encourage a small table and/or different sentences to remote the results for the indirect, direct and alternate visits. This is quite difficult to follow and understand.

Discussion

L274: This figure is confusing and misleading. Given the development nature of the work presented in the manuscript, the image and text for Figure 7 should be specified as juvenile fish. The authors have stated in L257 that their test didn't capture working memory in zebrafish yet this figure does imply that zebrafish do not have working memory and are not capable of behavioural consistency. I'd suggest to correct the figure to make the message more consistent with lack of findings for zf, instead of the direction shown currently. In addition to better labels that clearly identify the indices as variables in the current study only, consider less generalization of the weaker findings and instead focusing on clearer differences and stronger findings. Spelling of anxiolytic should be corrected.

L277-302: This paragraph is long and difficult to read. The writing itself could be improved but the argument about social buffering requires more support [citation and more specificity in description]. Only paradise fish are discussed without comparison or reference to zebrafish.

L304: Please provide a range of velocity and figure reference and/or citation for the slower velocity of the two species. Along with swimming velocity, please also state the size of the juveniles [mm, mean and range], particularly in reference to the testing arena [i.e., how much bigger were the arenas for zf and pf? 10X, 20X, 30X body length?].

L327: Please define what is meant by R-strategist.

L328-329: This is incorrect particularly for younger and smaller larval zebrafish and I encourage the authors to rephrase their claims in more accurate words or use the reference of adult fish instead of a larval fish. Various studies of ontogeny of shoaling and social preference show that first two weeks of life juvenile fish do not form 'shoals'. While juvenile fish progressively reduce distances between them and at pfd21 social preference can be measured, this behaviour is not quite as robust or motivating as the adult form. To the best of my knowledge 'returning to their shoal' and indeed forming tight shoals [only 2-3 body length apart as adult zf do] has not been yet reported at pfd21 [please include citations if this exists for zebrafish as it would be fantastic to read].

L335: 'All the features' is problematic here. Please consider listing some features instead of generalizing to all the features as a large body of neuroscience literature is focused on social behaviours and modeling social deficits which may be less apparent or absent in paradise fish.

L342: In general, the discussion section wasn't as strong as the rest of the manuscript and seems rushed. Please expand the discussion with commonly-expected paragraphs of comparison to existing pf and zf studies, limitations, future studies, comparison to other precocial fish species, etc. Please consider using more of the tone of 'complement' instead of 'more useful' or better. Use and establishment of paradise fish and other smaller fish species is great for understanding and comparing zebrafish [and social vertebrates] and vice versa. Translational work using both species would likely allow better understanding of evolutionary value, mechanisms, and pharmacogenetic regulation of social and non-social behaviours. The earlier parts of the discussion shy away from this notion and could be strengthened to better support that.

Materials and Methods

L347-353: As the factors mentioned below can affect both social and non-social behaviours of fish, please include how the fish were bred, if and how parental pairs were related, stocking density of parents, and housing density for embryos, juvenile, and older ages, and whether fish were housed in recirculation racks or glass tanks. Please also indicate housing water parameters [pH, salinity, ammonia/nitrates, etc.]. Please add this information as all these factors have been reported to affect behaviour of zebrafish [and possibly affect paradise fish] and help with interpretation of the results.

L353: Please state the dosage of anesthesia used for different ages.

L359-365: Please indicate the lighting conditions and feeding of the fish when they were in the opaque plastic tanks. Dimension of the isolated tanks are given but not of the social tanks; please state these as well. Please also indicate the group size of socially reared fish across development.

L371: Please provide some motivation for why such relatively high dosages were used for buspirone. Given that high doses of buspirone can cause drowsiness and movement abnormalities, what were the expected consequences for the drug treatments for anxiety behaviour that is recorded based on movement?

L375 and L394: The mention of Slalom test under the Sociability test [and of SJ and OT under the plus-maze] are a little confusing. Please consider either removing the redundant introductory sentences or having 'social tasks' and 'anxiety tasks' subheading at L375 and L394 before the specific details about apparatus and procedures for each test.

L379-380: Please indicate in text [and Figure 1A] whether the habituation was done with opaque dividers [as per the Dreosti protocol; <https://doi.org/10.3389/fncir.2015.00039>] or if the habituation differed in that aspect. Also, please state if fish were tested in sequential trials or several simultaneously multiple fish were tested? If so, how many at once? And could the test fish see other fish of same or different species? Could the test fish see the researcher placing stimuli fish at 15-min mark? If so, was something put in the empty end as well? Figure 1A also show the borders of social zone to be larger than Dreosti protocols [Figure 1 and 2 in Dreosti et al 2015], please clarify if these modifications were made or correct the figure to indicate the social and non-social zones more accurately.

L382: Please indicate what 'each zone' means in this context. Please also include any exclusion criteria here [i.e., freezing a significant amount of time or not ever going into a social and non-social zone, etc.].

L393: Please include any exclusion criteria here for the Slalom test. Same for L403, L409, L413, and L420.

L394: Please state the dimension of your plus-maze apparatus and the height of water-column for the center zone, and the

deep and shallow arms.

L410: Please indicate the size of the diameter of the well/arena.

L423 and L432: Please state whether all tested animals were socially reared for Experiment 1 and 2.

L433: Is the 8 referring to 8 dpf here? Also, why were 8dpf zebrafish not tested?

L437-8: What was the group size of house-mates for social housing?

L444 and L451: Please state the age of the zebrafish. Were zf also 30 dpf?

L447: Please state the age of both fish species.

Small Typos/errors

There are some trivial grammatical mistakes, unnecessary adverbs and colloquial terms, typos, missing commas etc. that make text difficult and informal and/or change the message of the sentences occasionally. Below are some examples, not an exhaustive list. Please ensure these and similar errors are corrected throughout the manuscript.

L23: Intra- and "interest" should be corrected.

L55-57: Use of the future tense "will" should be corrected. Ensure consistent spelling of behaviour throughout the manuscript.

L58 and L279: While not incorrect, please consider using a less colloquial terms instead of 'exactly', 'despite the fact', or removing these. Also, the question should be phrase 'what can we learn' in L58.

L62: 'Currently' should be 'recently'.

L66 and L280: The 'or' should be an 'and' in the listing of behaviours.

L67: Please check spelling; soliters should be 'solitary'.

L108: 'snow' should be 'show'.

L113-6, L277-279, L290-302: Use of the present tense should be corrected to past tense for existing literature and observed results.

L270-1: Spelling of showed.

L288: Missing linking verb 'is'.

Reviewer #2

(Remarks to the Author)

I highly recommend this article for publication. It is a well-executed and original study that underscores the importance of considering the natural history of a species when designing and conducting assays for exploration, anxiety, and working memory in fish. The research compares the sociality of juvenile zebrafish with that of paradise fish, concluding that zebrafish are better suited for social assays, whereas paradise fish are superior models for non-social novelty tests. The study provides compelling evidence supporting the use of paradise fish as a complementary model in biomedical sciences, particularly for individual-subject-based assays of exploration, anxiety, and working memory that require isolation—a significant confounding factor for other mainstream behavioral models.

I have only minor comments, which I believe will improve the article's readability and the replicability of its experiments.

Discussion

It would be valuable to discuss differences observed in adult fish across tests other than the U-shape. Additionally, consider how the findings of this study could inform future research.

In lines 23–25, it is mentioned that intra- and inter-test repeatability measures of the anxiety tests revealed that paradise fish express more consistent exploratory and defensive behaviors regarding time and context compared to zebrafish. However, this is not further discussed in the article, despite its importance for cognitive tests and for assessing personality traits.

In lines 295–302, it should be emphasized that longer periods of isolation need to be tested in future experiments. Social manipulation outcomes have been highly variable, with no consistent patterns emerging in zebrafish experiments (e.g., Buenhombre et al., 2021).

In line 303, while noting that paradise fish are more successful in novelty exploration across multiple paradigms, future studies should examine results with larger group sizes, such as a shoal, to enhance generalizability.

Results

It would be helpful to specify the experiment number under each heading to improve clarity.

Materials and Methods

The method of video recording is not described. Include details such as the camera used, and environmental conditions (e.g., illumination, background colors, distance from the camera).

Regarding environmental modifications, the composition of the groups is unclear. Specifically, how many animals were housed per group for the social condition prior to experiments 3 and 4? This information is crucial for replication, as stocking density significantly influences behavior (see Buenhombre et al., 2021).

In lines 359–365, it is unclear how many animals were assigned to isolation and how many to social housing.

Experimental Design

In lines 424–425, it appears that each "N" corresponds to each challenge, but this is not explicitly stated. Clarification is necessary.

In line 433, the context for each "N" is difficult to understand, particularly for paradise fish, where varying sample sizes are reported (e.g., n=21, 19, or 8; n=24, 30). Testing animals in pairs may introduce a confounding factor. For example, although zebrafish are social animals that form shoals, they exhibit increased aggression and dominance when in pairs (see Teles and Oliveira, 2016). This factor should be addressed in the discussion and results for the slalom test.

In line 437, it is unclear how many fish were isolated versus socially housed prior to the experiment. It seems that the first number in parentheses represents the "N" for social housing and the second for isolation, but this should be explicitly stated.

In lines 442–445, the concentrations of buspirone used are not detailed, though they are mentioned earlier. Specify which "N" corresponds to each dosage.

In lines 446–449, it is unclear whether the OT or the SPM experiment was conducted first. The sequence of experiments should be clarified.

Reviewer #3

(Remarks to the Author)

Title: Paradise fish (*Macropodus opercularis*) as a novel translational model for emotional and cognitive function

This study compares zebrafish and paradise fish juveniles in a series of sociality and novelty coping assays. The authors show that the paradise fish is less affected by presence or absence of social cues and is therefore a better model for studies where social isolation and individual tests are needed. The study is well executed and the paper is well written. But I have several comments and concerns about the assays and interpretation of results. Generally, there is also a major restructuring of sections needed. Please find below my specific comments.

Specific comments:

Lines 18 solitary instead of solitaire

Line 23 what is interest repeatability?

Lines 24-25 more consistent...regarding time and context... this is not clear. Please rephrase.

Lines 25-26 how can arm alteration test behavioural consistency? This tests flexible learning.

Lines 47-48 is till missing in this model

Line 67 function as solitary individuals.

Line 71 with or without the presence of conspecifics?

Line 75 zebrafish,

Lines 74-78 this should be dedicated to hypothesis or predictions, and not the result.

Lines 85-89 these lines actually belong to the analysis section and refer to what I write above in my comment.

Line 101 here and everywhere else, the no. of decimal places needs to be constant.

Line 108 show preference towards a conspecific in either condition.

Lines 112 that zebrafish lost its sociability is not agreeable. The results might be statistically non-significant, but the effect size may be still large. It is clear from the graph 1C that the zebrafish continue to prefer going to the conspecifics much more than the heterospecific or empty chamber. This is well established from previous research on shoaling preferences in this species, and anything otherwise would be strange. I would advise to consider the effect size for all tests and re-evaluate these results.

There is also a lot of interindividual variation in zebrafish (fig 1C), and there are a lot of fish that did not move or were inert. This kind of indicates that they did not like the setup or were shy or both. This must be accounted for. Or is this because of the darkness in Zantiks experiment chambers? I say this from personal experience in working with Zantiks.

Line 113 what is active aversion from a passive state?

Line 117 is not biased by the presence of

Lines 118 -124 these sentences do not belong to results section. They are methods and analysis.

In figure 2B, units are missing.

Line 133 mixed modelling

Line 138 what is meant here by exploration being more effective?

Lines 141-152 again, these belong to the methods.

In figure 3B, C units are missing.

Lines 155 F values should be written as F_{3,139}

Lines 167-171, 179-182 it is advisable to have a table instead of listing all the stats etc. here.

Lines 172-175 these belong to the discussion.

Lines 202 paradise fish is not affected by acute social isolation.

Lines 212-224 this whole paragraph belongs to methods section. Also, certain things are redundant as they have already been written in the methods.

Line 222 this should include inter and intraindividual variability. Is that considered?

Line 225 this is not clear, please rephrase.

Lines 224-232 the authors must include a table with all these results, within and between test variability, R measures, CI and P values. It is hard to understand these results in the way they have been presented at the moment. Similar holds for the correlation tests.

Figure 5D estimates of repeatability, whether significant or not, would decide if the behaviour is significantly repeatable. At the moment, it is hard to understand, given that these estimates have not been tabulated. A non-overlap of the bars would imply that the R estimates are significantly different from each other.

Lines 247-251 these belong to the methods. Moreover, can you confirm if the hypothesis is that 'fish that make more alterations will be considered to have better working memory'?

Lines 251-255 The velocity differences are not talked about here.

Generally, all this should be presented as a table. I would advise summarizing the results of all the behavioural tests in a comprehensive table.

Line 256 which exceed alterations of zebrafish (x %).

Lines 257-259 should go to the discussion.

Figure 6C legend and x axis labelling present redundant information. Further, the observation that zebrafish had higher velocity despite making less visits, revisits and alterations indicate that the fish could be stressed and thus made random movements.

Lines 270-271 show and not showe

Lines 275-276 the figure description is insufficient to explain how effect sizes contribute and what the figure means. A detailed description is needed. Anxilytic should be replaced by anxiolytic.

Lines 277-279 adult and juveniles are ought to differ in behaviour. And the explanation here belongs to introduction where the authors must explain why they did this study on juveniles rather than on adults.

Lines 290-293 this can be instead interpreted as social inquisitiveness, with paradise fish making more inspections than zebrafish.

Lines 294-295 there is no predator here and so I doubt this can be related to that. The zebrafish seem to be smaller in size than the paradise fish and so it cannot be a predator.

Line 302 presence or absence of conspecifics

Lines 314-319 as mentioned above, and as reiterated by the authors here and in the next paragraph, the y maze movement pattern tests for exploration and not working memory. I have concerns about it being used to assess working memory. The fact that zebrafish use more indirect visits probably show indicate that it is more random in its behaviour.

Lines 326-328 this sentence is not clear. What is a R strategist?

Lines 332 the juvenile zebrafish and paradise fish are different in body size. I think this difference itself can contribute to differences in behaviour. The authors must discuss this.

Lines 348-349 zebrafish between 10 and 30 days are not just juveniles. They are larvae around till 20-25 days beyond which they become juveniles. Did you consider that change?

Line 354 terminated with tricaine?

Line 384 why did you use larval and juvenile fish? This is not clear and should be explained in the introduction. Larvae and juveniles can be differently active, are much less social than adults and this can be a potential confound. For example, see studies by Roy and Arlinghaus 2022, Sbragaglia et al. 2021, 2022 where they measured ontogenetic change in collective behaviour in zebrafish.

Line 414 the working memory test is not clear. How do you test working memory actually in these fish? If it is using the FYMP, then I am not sure how much learning and memory is involved here. It has a strong confound with fish exploratory behaviour, and alteration or direct or indirect t revisits can be easily interpreted as exploration. I am aware that there are previous studies in mice that have used this since the method is developed using the Zantiks. But I would be highly conservative to regard this as working memory and not as mere exploratory behaviour. And the results resonate my observation above (ref to lines 255-259 in the main text).

Line 446 why did you have a separate test for behavioural consistency? It could have been tested with the tests that you did previously like open tank, surface avoidance, exploratory test, etc.

Line 447 'used' instead of 'applied' zebrafish.

Line 458 stats was done using R (ref.).

Lines 457-478 you write what tests were done. But you should describe the analysis by saying what comparisons you did, what were the different variables used, what was the formula for each kind of test, what kind of transformation was done for the data, etc.

Version 1:

Reviewer comments:

Reviewer #1

(Remarks to the Author)

I am thankful to the authors for addressing my concerns and questions and am satisfied with their answers and changes.

Dear *Communications Biology* Editors,

We would like to thank the opportunity to resubmit our revised work on highlighting the potential advantages of using the paradise fish in translational research. We are also extremely grateful for all three reviewers for their insightful suggestions to enhance the accuracy, readability and relevance of our work. Their overall positive feedback and supportive comments have been very motivating for us during the revision process and as you will see in our itemized answers below, we have addressed all their comments.

We hope that both you and the reviewers will find our answers satisfactory.

Best regards,

Máté Varga

Itemized answers for the Reviewers' comments:

Reviewer #1 (Remarks to the Author):

This paper by Varga et al was a stimulating read. The authors report interesting and novel behavioural data highlighting differences between young zebrafish and paradise fish to establish and further demonstrate utility of paradise fish in research. They considered three main behavioural contexts: exploration, anxiety, and working memory, that may provide potential of modelling for various neuropsychiatric disorders and basic neuroscience research using paradise fish. Species-specific differences in social interaction, non-social exploration, aversion/desirability to novelty, and responses to anxiety and pharmaceutical anxiolytic agents are important scientific investigations, for both translational neuroscience researchers and broader academic community. Using individuals and paired fish for same tasks and designing novel tasks [e.g., Slalom test] is exciting and commendable. The focus on two developmental stages is also a strength of the manuscript though that comes with caveats of limited ability to make inference about any effect of sex and about the mature forms of behaviours in the two species. I enjoyed reading the manuscript and applaud the authors for their good work. I have some more serious queries and comments, along with minor and easily fixable suggestions emphasised below. Please considering these clarifications and changes as I believe these can enhance confidence in the study's findings.

We really appreciate both the positive review and constructive comments of Reviewer 1. As detailed below, we have addressed the flagged issues and implemented the requested revisions accordingly.

General Issues

As is the case in such studies, all of the behaviours/variables reported are movement-based, which can be affected by the relative size of the fish, for both within and inter-species

comparisons. There can be large variation between siblings raised together and isolated fish can be larger within the first 3 weeks of life as they do not face much competition for food whereas social rearing leads to some monopolizing of food by some siblings. It would be really helpful to have some indication [mean and/or range of body length] of the size of both species as the same arenas are used for all fish of the two species and the same dimensions may represent different level of difficulty [e.g. arena size of 20X body length may induce different level of anxiety than 15X]. Potentially some of the variation could be due to size differences between zebrafish and paradise fish [and social vs. isolated fish of same species; and within same treatment siblings as well]. Please include a figure or table for size of fish; it would help with interpretation of results.

Thank you for highlighting this important potential confounding factor in our research design. We agree that physical differences such as body size can influence and in some cases limit behaviour. To investigate this as a potential interfering factor, we measured body length on the video recordings of the social isolation experiment of both species. We chose this experiment to make the measurements because social isolation can also bias development and body size. We added the body length data and also centrum entries (locomotion) data to the relevant figure (Figure 3B and 3C). Briefly, we found that juvenile paradise fish are shorter compared to zebrafish of the same age, and this is independent of rearing condition (isolation vs. social housing) during the studied developmental window.

Given these species-specific differences in body size, we directly compared locomotion, general exploration and stimulus exploration in three tests that target different domains: sociability (U-shape test), anxiety-like behavior (SPM) or cognitive performance (Y-maze). We represent locomotion, general exploration and stimulus exploration with mean velocity, intersection entries and social entries in the U-shape test; mean velocity, centrum entries and shallow arm entries in the SPM test, and mean velocity, all entries and alternated entries in the Y-maze test in Supplementary Figure 2. Across all paradigms, paradise fish showed either comparable or lower locomotion relative to zebrafish. In contrast, both general exploration and stimulus-directed exploration was consistently elevated, regardless of the type of the behavioral domain tested.

Paradise fish consistently explore areas more efficiently, although they do this at a lower speed. This indicates that differences in body size and reduced locomotion do not limit the performance of these animals in these tests. Notably, in the U-shape sociability test paradise fish, increase their swimming speed in the presence of zebrafish, matching the velocity typically observed in zebrafish under the same conditions. This also demonstrates that paradise fish would be capable of reaching higher swimming speed, but they generally choose a more deliberate exploration strategy. Collectively, these results indicate a substantially different exploration strategy between the two species. We also want to note that in the majority of experiments, we did not compare the two species directly, but comparisons were made to their respective conspecific controls and the results reported if the given observed factor (like social company or a medical treatment) biased the behaviour. This within-species experimental design minimizes confounding effects arising from the different motor abilities of the two species.

We added some discussion of this potential confounding factor in the Discussion section of the revised manuscript (lines 386-396).

The reporting of findings and later discussion would also benefit from reduction in generalization and toning down some of the less supported claims. While the authors focus on the dichotomy between the two species, in parts of the manuscript and in similar literature, fish [and other animals] can respond to anxiogenic/anxiolytic factors by increasing OR decreasing locomotion and even both in different ways within a short time [e.g. freezing with bouts of fast/erratic movements]. A more comprehensive understanding would be possible for the reader

if locomotion is not defined as an axis but rather a grid or plane where the same treatment can affect both hyperactivity and freezing. To this end, while I appreciate the authors providing composite score for anxiety, exploration, etc., please consider providing simpler movement [distance travelled/velocity, or distance from the anxiogenic stimulus/side] over the whole time of testing as supplementary information in this and future manuscripts to aid with interpretation of the results. I think this would make the findings more interesting and reinforced.

We thank the Reviewer for this advice. In the revised manuscript we have included locomotion-based variables, specifically velocity, centrum enter frequency, to all experiments in the main text. Source variables of anxiety scores in the case of the anxiety experiment are now provided as a Supplementary Figure 1.

Abstract

L24: Given lack of aggression testing in the manuscript, I would encourage authors to rephrase 'defensive behaviours' to a more precise term regarding risk-avoidance in novel environment or protection from predator in their behavioural testing.

Thanks for this suggestion, we have changed the phrasing to risk-avoidance, as recommended.

Introduction

L49: The authors use the phrase 'such a simple social structure' to define natural shoaling tendency of zebrafish. I find that difficult to follow as shoaling is anything but simple. With competition for resources, dominance hierarchy, disease, and other psychological and health dynamics, etc., some are mentioned by the authors in the next sentences, shoaling can be quite complicated. Could they elaborate on this and/or provide a more suitable phrase. Also, it would be useful to know if their 'simple' designation of zebrafish shoaling is in comparison to solitary lifestyle or other more/less interactive social structures.

In the original manuscript we referred to natural shoaling as “simple” as it is unconditional, permanent and almost reflexive, and it does not require complex social decisions, unlike the courtship or territorial behaviours of paradise fish. However, we agree with the reviewer that while shoaling behaviour might seem less complex than the social behaviours of paradise fish, ultimately it is not simple at all, so we have removed this phrasing from the revised manuscript.

L63: Again, please use phrasing that is more accurately representative of antipredator behaviour. 'Defensive' the broader term used here, implies both conspecific aggression/competition and protection from predators and other harm.

Thanks for this suggestion, we have changed the phrasing to risk-avoidance, as recommended.

L65: Please provide more support here for 'one that resembles more human social settings'. Do the authors imply this only for more individualistic societies? What specific social behaviours or dynamics make paradise fish more similar to human socialization. As one could also argue the opposite that more social species like zebrafish bear more similarity to humans, particularly collectivistic cultures so such an elaboration here [1-2 sentences] would be interesting and provide a more comprehensive understanding for the readers.

What we intended to convey here is that despite zebrafish, paradise fish and humans all being social species, the structure and cohesion of their social groups, and consequently their species-specific social behavioural repertoire differs markedly. Zebrafish, and shoaling fishes in general, live their life almost permanently as part of the shoal. This minimizes the need to perform higher order social functions, like context-dependent conflict resolution (assessing conspecifics to decide to fight them or not), individual

recognition, choose mating partners, or active parental care. They are part of a (seemingly) homogenous group where social bonding is unconditional and communication signals are generalized rather than directed toward specific individuals. In contrast, in paradise fish or, indeed, in humans, social behaviour is more diverse and depends on the context. Paradise fish choose their partners, build their foam nest, defend their territory, and display parental care. Such behaviours require decision making and plasticity. Due to this context-dependent nature of their social behaviour, we hypothesise that both humans and paradise fish can function both within and outside of their social group, making paradise fish a more suitable model for studying responses to novelty challenges in individual-based behavioural tests (lines 70-80).

L66: What does function imply here as various important life necessities are already listed earlier as social activities. Please list elements of life that are performed individually by paradise fish. It would also be helpful to know if paradise fish show any parental behaviour [nest building and guarding embryos and young] and for how long. Can they be classified as precocial or altricial? When do young paradise fish become solitary? Brief mentions of these behaviours would help with better comprehension of the mentioned 'challenges' in the next sentence as well.

Thank you for raising this important issue. Male paradise fish express nest building behaviour and the territorial defensive behaviour as solitaires. Usually, nest building takes about 2-3 days. Males also express agonistic interactions, such as courtship behaviour and parental behaviour, and the mean duration of parental care (mouthing, bubbling, and nest maintenance) was 5-6 days (Ward, 1967 *Copeia* <https://doi.org/10.2307/1441891>). Young paradise fish start to hatch from the chorion at the end of the second day, and just as zebrafish after leaving the chorion, are precocial (lines 73-80). Larval paradise fish are not strictly solitary and the transition towards solitary behaviour is a gradual process, with the first signs of increased aggression appearing in the juvenile stage and becoming more pronounced during the adolescent stage

L76: Could you describe here what is meant by 'human phenomena'?

In this part, "human phenomena" refers to those complex functions that we aim to model with these animals, such as anxiety or cognition. As an effort to decrease the tone and avoid overinterpretation we excluded this part from the revised manuscript

Results

L86-89: This description could be more helpful if it mimics the same order as the Figure 1A-C [Cons, I-Spc, Both].

Thank you for this suggestion, we have revised the text accordingly.

L90: Figure 1B would allow better comparison if the y-axis is the same for the two graphs. Is the obvious size difference in the images of zebrafish and paradise fish on the top of Figure 1B and 1C [and later figures as well] intentional and represented of tested fish? As mentioned in the comment above, size of the fish can affect many of the tested behaviour in the same arenas, please elaborate if there was a difference between the two species in size at pfd8 and pfd21. Please also comment on the variability in size within each species at these ages.

We have changed the y axis for the respective panels in the revised manuscript. While the age of the tested fish is always stated, for the sake of clarity we always used pictograms of adult animals. We measured the sizes of the tested fish and this information is now represented in the social isolation experiment for juvenile fish.

L90-Figure 1C: It is difficult to draw conclusions about sociability from the ‘% time in chamber’ variable alone. It is common to use the time values for the two zones in this task and calculate before/after social preference index [SPI] as Dreosti et al [2015; <https://doi.org/10.3389/fncir.2015.00039>] did and others since with this type of protocol and data. I am also concerned that some of the fish spent 0 or 100% of time in one or the other zone. Were these fish frozen/immobile in a zone or did they explore the whole testing area at some point to have awareness of other chambers? These graphs are difficult to decipher without the comparison of habituation vs post-social stimuli and if the exclusion criteria don't include exploration of the whole testing area. Please include more details on this in the manuscript.

Thanks for pointing this out. In the revised manuscript we have added the time (%) spent and number of entries to the intersection zone (see revised Figure 1) so this way the percentages cover the whole testing arena. Unfortunately, we cannot provide data from the habituation, as these sessions were not recorded. However, we have extensive prior experience with the test, and during the validation process we recorded many sessions of habituation in zebrafish at 8, 14 and 30 dpf, in control conditions or following social isolation (PMID: 32350040). We did not find any a priori spatial bias towards any of the chambers or directions. Neither the age nor the way of rearing seemed to influence the exploration of the empty test apparatus. Based on these findings, we discontinued the video-recording of the 15 minute-s habituation phase in subsequent experiments. Similarly to the rodent sociability paradigm, we instead use the presence of the social preference (above 50% in social zone) and adequate locomotion as quality-control criteria in this test, which are both met in our current experiment. We also note that while habituation data was presented in the aforementioned Dreosti paper, which introduced the test for the first time (PMID: 26347614), it was not presented in the follow-up papers where the test was used to compare the effect of isolation to control, or the rescue effect of buspirone (PMID: 32366356).

Swimming velocity data and intersection crossings data is also provided now, which show that the animals do not express freezing behaviour during social challenges. The approximate 4 mm/sec swimming speed of the zebrafish is in line with the speed measured in the anxiety tests and the y-maze test in this study (see Supplementary Figure 2). We cannot exclude the possibility that a few animals stay in one chamber, but the data indicates strong social preference, proper exploration and no sign of freezing.

L90-99: Please indicate the age of the fish on the figure and the caption. I recommend being consistent with terms for the arena parts. It is not clear if ‘zone’ is the same as ‘chamber’ or if they represent different parts. Figure 1B is difficult to interpret as the left panels lacks any indication [or ns] shown in 1C. Is the velocity for the whole 30-minutes or just the 15-min habituation, or the 15-min of the social test phase?*

Thank you for pointing out the inconsistency of our initial phrasing. We now use the term “zone” consistently instead of zone or chamber in the revised manuscript. We have also added the age of the animals in every result section and figure legends. We have also added all indications of significant and non-significant results. The velocity was measured during the 15 minutes social phase of the test.

*L96-97: It may be helpful to state that * represents significant difference from Conspecific condition [C-spc]. The current phrasing makes the use of * difficult to follow here.*

In the revised version of our manuscript, we make this clear in the figure legends. We also made an effort to indicate significance in a more straightforward way on the plots.

L97-99: In the figure caption, please indicate if the values in Figure 1C represent time during the second half of the test or the whole test? Please consider using ‘social zone’ instead of chamber in figure 1C.

Thank you for the suggestion. These values refer to the second half of the observation period, when we present the stimulus fish (we consider this the test period). We tried our best to improve the clarity and coherence of the figure, its legend and the corresponding parts of the main text.

L100-103: The graphs in Figure 1B show higher values and variation for zebrafish than paradise fish, which is not given much consideration at all in the manuscript until a brief mention in the discussion. Please state expected range of swimming speeds for the two species and include if young zebrafish have been reported to swim faster than paradise fish in literature or if this is just a unique finding or artifact in your study.

To the best of our knowledge, no previous studies have directly compared the swimming speeds of zebrafish and paradise fish. This gap likely reflects several factors. Primarily, swimming speed comparisons are generally of limited scientific value on their own and are typically only relevant in contexts where velocity differences may act as potential confounding variables - such as in the present study. As this is the first study to systematically compare these two species within a preclinical framework, it is also the first instance where such a comparison becomes necessary. Attempting to draw comparisons from existing literature that focuses on only paradise fish is problematic, as most of those studies do not report swimming velocity data. Swimming speed measurements have only become common with the broader adoption of automated video-tracking systems (e.g., EthoVision, Viewpoint). Such automated measurement systems have only recently become standard tools in behavioural neuroscience, long after paradise fish peaked in the field of ethology.

In the revised manuscript, we provide swimming speed data for both species across all relevant experimental contexts. Velocity measures are reported in every experiment where they were collected, and a direct comparison between species is now included in Supplementary Figure 2.

We can only speculate on higher variability of sociability in paradise fish compared to zebrafish. As noted previously, the social behaviour of paradise fish appears to be more conditional, which may manifest as increased behavioural plasticity and variability in social contexts. While the social motivation of zebrafish in choice paradigms is relatively straightforward - primarily driven by a desire to reunite with their group - the social motivations behind paradise fish behaviour are likely to be more complex and may include both affiliative and antagonistic components.

L111-116: The interpretation portions of this summary [lost preference, developed aversion, etc.] would be better suitable in the discussion. Please only summarize the data trends here.

We rephrased the relevant parts of the manuscript and removed these interpretations as suggested.

L118-119: This sentence needs citations. Please cite existing studies that first established and have since demonstrated social buffering in zebrafish and other social species.

We have added citations referring to the first description of social buffering in zebrafish, a review with multiple fish examples, another review with vertebrate examples, as well as the original papers reporting social buffering in cichlids, minnows, tilapias, koi carps, daces, and sturgeons (lines 149-150). We also put more focus on social buffering in the Discussion section.

L123: Please indicate what age is meant by larval zebrafish and juvenile here.

In the revised manuscript we use the exact age instead of vague terms, such as “larval” and “juvenile”. Additionally to naming the exact age, we rephrased larval and juvenile to “early larval” and “late larval” period, respectively, to refer to these developmental stages more specifically in the revised manuscript.

L128-131: The interpretation parts of this sentence should be avoided in the results section and be placed in the discussion.

We replaced it in the revised manuscript accordingly.

L134: BIC should be written out as bayesian information criterion and briefly defined at first use.

We corrected this in the revised version of the manuscript (line 164).

L139: Figure 2D would allow better comparison if the y-axis is the same for the graphs. In general, lack of data from pfd8/larval paradise fish makes the pfd8 zebrafish data less useful, and the figure more distracting and complicated. Larval zebrafish are also much smaller at pfd8 [compared to pfd21] and the same testing arena may have presented a much greater challenge for pfd8 zebrafish than pfd21 zf and pf. I'd encourage the authors to take out pfd8/larval zf data from this figure and analysis, and focus only on the pfd21 zf and pf comparison.

Our original aim with this figure was to illustrate how exploratory dynamics differ between single and paired individuals across the examined age/species groups. Since the central focus is on comparing these conditions, we chose to standardize and maximize the y-axis scale for clarity and direct comparability within these. While 8 dpf zebrafish are indeed smaller in size, their success rate as individual explorers in the slalom test is not significantly lower: 60% of 8 dpf fish reached the final chamber compared to 63% of 30 dpf zebrafish. Importantly, the 8 dpf zebrafish were initially introduced as a non-social control group. Thus, if differences between paradise fish and zebrafish were driven solely by sociality, one would expect paradise fish to perform similarly to 8 dpf zebrafish. However, the three groups display distinct exploration dynamics, indicating that fundamental species-level differences in exploration strategies exist, that cannot be fully explained by social motivation alone. Given the additional information provided by the early larval zebrafish group, we believe this dataset adds valuable context and therefore prefer to keep it in the manuscript.

L142: Please provide some motivation here for why this specific amount/level of isolation [3 days] was chosen; as compared to more commonly used 1 hour, 24 hours, 7/14/21 days that have been previously used by the authors [Ref 24] and many others.

This isolation protocol builds upon the commonly used 24-hour isolation paradigms in fish (see PMID: 29189020) by incorporating additional sampling points every 24 hours across a full 3-day period. Extending the duration of isolation allows for a more representative model of chronic social isolation, while the high-resolution, day-by-day sampling enables us to track the development of behavioural effects over time. This design not only captures the dynamics of isolation-induced changes but also identifies a potential time window during which paradise fish can be used in multi-day experimental protocols while maintained in isolation.

L153-160: These results are interesting as these are in contrast of published reports of higher exploration and activity in open-field type testing following isolation of young zebrafish. Did the authors find any difference in overall locomotion [not exploration but mobility level] of fish after isolation?

In our previous work (PMID: 32350040) with isolation-reared zebrafish, we have not observed increased locomotor activity. Despite our extensive experience with this model, both exploration and general locomotion have typically been reduced or unchanged following social isolation, as shown in our earlier studies using early and late larval stages (PMID: 32350040). Reports of enhanced locomotion are generally linked to chronic isolation protocols, typically applied to adult fish and/or initiated at the embryonic stage - both of which differ substantially from the current model. Furthermore, systematic comparisons of isolation paradigms, such as those by Shams et al. (PMID: 29091281), also demonstrate

that peri-metamorphic (approximately 30 dpf) isolation leads to reduced locomotion. In the current study, which employed a brief isolation protocol, we observed no difference in general exploratory behaviour (e.g., centre entries in the swimming plus maze; see Figure 3C), but a reduction in risk-associated exploration across both anxiety tests in zebrafish. Unfortunately, swimming speed was not recorded in this experiment. We compare our results with results from the existing literature of social isolation in the revised discussion paragraph.

L166-169: Figure 3 is difficult to see due to the size of font [even if one can ignore the line numbers superimposed on top of the right side of the figure where significance is indicated with vertical bars]. Please consider changing the size and orientation of this figure for better visibility and briefly summarizing the significant findings, trends, and general direction of the results in the caption.

Thanks for this suggestion. We have revised the figure to improve clarity, including increasing font size and enhancing readability. Additionally, we added summary text to help guide interpretation. Most of the detailed statistical results have been moved to the accompanying Supplementary Tables in the revised manuscript, which we believe improves overall readability and makes the findings easier to follow.

L182: Consistent with previous two sub-sections in the results, this sub-section would benefit from a summary sentence at the end of this analysis.

As suggested, we added summary sentences in all sections in the revised manuscript.

L193-202: These results are interesting and I'd encourage the authors to keep the interpretation elements for the discussion. Generally, the reporting of findings and later discussion can also benefit from less generalization.

In agreement with the suggestion, we tried to decrease the tone of the statements generally, and also moved interpretation parts from the results to the discussion section.

L203-Figure 4B and 4C. The colour combinations and y-axes are confusing here. Please use the same range for y-axis for B and C. Consider using different combinations of colours to indicate isolation treatment [e.g., grey & teal used in Figure 3] and buspirone treatment. It is not clear just by looking at the figure 4C which fish were isolated and which were not.

We apologise for the use of confusing plots. In the revised figures we use uniformly the same color code as we did in the previous results section. Figure 4B does not show results from a new experiment but shows a set of results from the previous social isolation experiment to compare that effect with the anxiolytic treatment (it appears in the main text at line 228). We added all the source variables of the isolation experiment and the buspirone experiment to the Supplementary Figure 1.

L221: Where were fish placed during the ITI? Home-tanks or new tanks?

The subjects were kept in the same test apparatus for assessing within-test variability and recorded sessions in visible light were separated by 5-minute dark periods during the ITI (line 558-559). Following this, animals were transferred to the other test apparatus where they were exposed to the same procedure to assess between-test variability. The order of the test types was previously randomized.

L236: Figure 5C is difficult to see. Please make the grey bars and labels darker and more visible.

Thank you for pointing this out, we enhanced the contrast of the figure by making bars and labels darker. We also report the repeatability values and confidence intervals in Supplementary Table 5.

L252-255: I'd encourage a small table and/or different sentences to remote the results for the indirect, direct and alternate visits. This is quite difficult to follow and understand.

We moved all statistical test results into Supplementary Tables now. The above-mentioned differences are also included in this table, which, we hope, will increase clarity.

Discussion

L274: This figure is confusing and misleading. Given the development nature of the work presented in the manuscript, the image and text for Figure 7 should be specified as juvenile fish. The authors have stated in L257 that their test didn't capture working memory in zebrafish yet this figure does imply that zebrafish do not have working memory and are not capable of behavioural consistency. I'd suggest to correct the figure to make the message more consistent with lack of findings for zf, instead of the direction shown currently. In addition to better labels that clearly identify the indices as variables in the current study only, consider less generalization of the weaker findings and instead focusing on clearer differences and stronger findings. Spelling of anxiolytic should be corrected.

Thank you for pointing out the potentially misleading nature of the summary figure. In the revised manuscript, we have updated the figure title (to “Sensitivity of behavioural paradigms applying paradise fish or zebrafish or late larval stage (30 dpf)”), so it now better reflects both the function and the representativity of the figure. Our primary aim with this figure was to provide a visual guide and illustrate which species are more suitable for existing fish paradigms used to assess social, exploratory, or cognitive traits. This figure is not intended, therefore, to imply the presence or absence of these traits in a given species. We argue that despite paradise fish being a social species, their sociability cannot be measured with a U-shape test, and despite zebrafish having working memory, their explorative strategy is less suitable to measure this in the y-maze test, just to mention the two strongest discrepancies what we measured in the current study. However, we do understand that in its previous form the plot gave the impression of listing traits of the species. We have also revised the figure legend to include a detailed explanation of how the presented values were calculated (lines 318-324). Additionally, the previously mentioned typographical error has been corrected.

L277-302: This paragraph is long and difficult to read. The writing itself could be improved but the argument about social buffering requires more support [citation and more specificity in description]. Only paradise fish are discussed without comparison or reference to zebrafish.

We thoroughly revised the paragraph in the updated manuscript, placing particular emphasis on providing more specific, well-supported background information and including cited examples of social buffering (lines 348-357).

L304: Please provide a range of velocity and figure reference and/or citation for the slower velocity of the two species. Along with swimming velocity, please also state the size of the juveniles [mm, mean and range], particularly in reference to the testing arena [i.e., how much bigger were the arenas for zf and pf? 10X, 20X, 30X body length?].

We added locomotor and exploratory variables to all experiments in which these data were collected, along with a summary plot provided in the supplementary materials (Supplementary Figure 2, Supplementary Table 7).

Additionally, we included body length measurements for Experiment 3 (Figure 3B), presenting boxplots that display medians, quartiles, and individual data points, as a function of species and rearing condition. Independently of social isolation, paradise fish larvae were approximately 2 mm in length, compared to

2.5 mm in zebrafish larvae. Based on the dimensions of the test apparatuses, this means that, to reach the farthest point from their starting position, paradise fish vs. zebrafish had to swim approximately: 42.5× vs. 34×, 16× vs. 12.8×, 10.5× vs. 8.4×, 21× vs. 16.8×, 17.5× vs. 14×, 15× vs. 12×, or 30× vs. 24× their body length in the U-shape, slalom, SPM, showjump, open-tank, or Y-maze tests, respectively.

However, as previously noted, the two species differ in their swimming mechanics and exploratory strategies. Our data indicate that exploration is neither limited nor determined by body size and, in fact, appears not to be correlated with it.

L327: Please define what is meant by R-strategist.

The “r-strategist” (or “r-selected species”) designation in ecology refers to species that are characterised by small size, quick reproduction time and short lifespan due to their fluctuating environment (see Jeschke 2008, <https://doi.org/10.1016/B978-0-12-409548-9.11121-2>). Zebrafish are generally described as opportunist animals whose quick reproduction cycle is associated with raining and the associated food surplus, hence they can be considered r-strategists. These species cope with environmental challenges with adaptation on the population level (producing a new generation of individuals in great numbers) instead of adaptation on the individual level (phenotypic plasticity). R-strategists are less likely to give plastic responses, as they rely on their innate behavioural repertoire.

We acknowledge, however, that using this term in the given context is too speculative and would require a more in-depth explanation. As such an argument is out of the scope of our manuscript, we removed it from the revised version.

L328-329: This is incorrect particularly for younger and smaller larval zebrafish and I encourage the authors to rephrase their claims in more accurate words or use the reference of adult fish instead of a larval fish. Various studies of ontogeny of shoaling and social preference show that first two weeks of life juvenile fish do not form ‘shoals’. While juvenile fish progressively reduce distances between them and at pfd21 social preference can be measured, this behaviour is not quite as robust or motivating as the adult form. To the best of my knowledge ‘returning to their shoal’ and indeed forming tight shoals [only 2-3 body length apart as adult zf do] has not been yet reported at pfd21 [please include citations if this exists for zebrafish as it would be fantastic to read].

We apologize the mistake and agree completely with the reviewer that the social groups formed by one month old zebrafish should not be considered shoals. We changed the phrasing throughout the revised manuscript.

L335: ‘All the features’ is problematic here. Please consider listing some features instead of generalizing to all the features as a large body of neuroscience literature is focused on social behaviours and modeling social deficits which may be less apparent or absent in paradise fish.

Thanks for this suggestion - we decreased generalisation in the respective part of the revised manuscript and specifically listed features such as company size and early behavioural development as shared advantageous features of paradise fish and zebrafish.

L342: In general, the discussion section wasn’t as strong as the rest of the manuscript and seems rushed. Please expand the discussion with commonly-expected paragraphs of comparison to existing pf and zf studies, limitations, future studies, comparison to other precocial fish species, etc. Please consider using more of the tone of ‘complement’ instead of ‘more useful’ or better. Use and establishment of paradise fish and other smaller fish species is great for understanding and comparing zebrafish [and social vertebrates] and vice versa. Translational work using both species would likely allow better understanding of evolutionary

value, mechanisms, and pharmacogenetic regulation of social and non-social behaviours. The earlier parts of the discussion shy away from this notion and could be strengthened to better support that.

We made a serious attempt to rewrite the discussion according to the suggestions. In the revised version of the manuscript all results are discussed and - where it is possible - compared to existing literature. A limitation section is also provided now. We agree that the paradise fish should not have a replacement but a complementary role in the field of preclinical neuroscience, hence we decreased the tone of our statements according to this.

Materials and Methods

L347-353: As the factors mentioned below can affect both social and non-social behaviours of fish, please include how the fish were bred, if and how parental pairs were related, stocking density of parents, and housing density for embryos, juvenile, and older ages, and whether fish were housed in recirculation racks or glass tanks. Please also indicate housing water parameters [pH, salinity, ammonia/nitrates, etc.]. Please add this information as all these factors have been reported to affect behaviour of zebrafish [and possibly affect paradise fish] and help with interpretation of the results.

Adult zebrafish in our facility are maintained in automated Tecniplast racks, under standard conditions: water temperature is 27.5°C, pH 7.2, conductance of 430 µS. Adult paradise fish were maintained as described before (PMID: 33799915), in glass tanks with weekly water changes. The conditions were: pH 6.8-7.5, conductance: 350-450 µS (the slight increase is due to evaporation), temperature: 25-27°C. Nitrite, nitrate and ammonia concentrations were measured regularly, and they were within the following limits: nitrite 0.05-0.08 mg/L, nitrate 10-15 mg/L, ammonia: less 0.05 mg/L. Fish larvae of both species are moved from the incubator to the facility at the age of 5 dpf, where they are kept in 2.5L containers. At the beginning stocking density is 40 fish/tank, at one month post fertilization we reduce the density to 20-25 fish/tank, and around two months of age to 10-12/tank. We have added now this information to the revised manuscript (lines 448-463).

L353: Please state the dosage of anesthesia used for different ages.

We clarified now that according to Hungarian regulation we used a 400 mg/L concentration for euthanasia with tricaine overdose (lines 461-462). (Note that this is significantly higher than the 164 mg/L concentration of tricaine that we use in the lab for larval anaesthesia, and 120 mg/L in juveniles fish.)

L359-365: Please indicate the lighting conditions and feeding of the fish when they were in the opaque plastic tanks. Dimension of the isolated tanks are given but not of the social tanks; please state these as well. Please also indicate the group size of socially reared fish across development.

Lighting and feeding details of fish is stated in the materials and methods section, Animals subsection (lines 446-464).

“Fish were maintained in a standard 14 h/10 h light/dark cycle. Feeding of zebrafish and paradise fish larvae started at 5 dpf with commercially available dry food (a 1:1 combination of <100 µm and 100-200µm Zebrafeed, Sparos). After 15 dpf, juvenile fish were fed using dry food with gradually increasing particle size (200-400µm Zebrafeed, Sparos) combined with fresh brine shrimp hatched in the facility.

The volume of social tanks was 2.5 L (16x16x12 cm), in which approximately 40 early-stage larvae were placed.

L371: Please provide some motivation for why such relatively high dosages were used for buspirone. Given that high doses of buspirone can cause drowsiness and movement abnormalities, what were the expected consequences for the drug treatments for anxiety behaviour that is recorded based on movement?

Although the concentrations of 25 and 50 mg/L buspirone are considerably higher than those typically used in rodent or human studies, they are the most commonly applied doses in zebrafish research. This dosage range was originally established by Maximino and has been repeatedly confirmed and revalidated by our group in both published and unpublished work (PMIDs: 30410116 and 32350040). Based on our experience, 50 mg/L buspirone is only sedative in early larval stages, while 100 mg/L is the lowest concentration that reliably induces sedation across all developmental stages.

In the current study, neither 25 nor 50 mg/L buspirone produced locomotor effects in zebrafish, as indicated by swimming velocity, exploration latency, and entry frequency. In contrast, paradise fish exhibited reduced swimming speed at both concentrations, suggesting a potential mild sedative effect. However, only the higher concentration significantly altered anxiety scores and their contributing variables. This dissociation implies that changes in locomotor and anxiety-related measures are not directly coupled and may reflect distinct underlying processes.

Notably, swimming speed in paradise fish was also modulated by social and heterospecific cues, supporting its potential role as an anxiety-sensitive variable in this species. We agree that further investigation of buspirone and other psychotropic agents is essential for validating the utility of paradise fish as a translational model.

L375 and L394: The mention of Slalom test under the Sociability test [and of SJ and OT under the plus-maze] are a little confusing. Please consider either removing the redundant introductory sentences or having 'social tasks' and 'anxiety tasks' subheading at L375 and L394 before the specific details about apparatus and procedures for each test.

As requested, we have added “social exploration tasks” and “novelty exploration tasks” subheadings and moved the introductory sentences below those to the chapter.

L379-380: Please indicate in text [and Figure 1A] whether the habituation was done with opaque dividers [as per the Dreosti protocol; <https://doi.org/10.3389/fncir.2015.00039>] or if the habituation differed in that aspect. Also, please state if fish were tested in sequential trials or several simultaneously multiple fish were tested? If so, how many at once? And could the test fish see other fish of same or different species? Could the test fish see the researcher placing stimuli fish at 15-min mark? If so, was something put in the empty end as well? Figure 1A also show the borders of social zone to be larger than Dreosti protocols [Figure 1 and 2 in Dreosti et al 2015], please clarify if these modifications were made or correct the figure to indicate the social and non-social zones more accurately.

We have added the required details to the “Behavioural procedures” and “Experimental design” subsection of the Materials and Methods section. In summary, all focal and stimulus fish were naive to the test. The same apparatus were used for the habituation and testing phases, made of white opaque material (VeroWhite PolyJet Resin). Six fish were tested simultaneously in separate apparatus.

L382: Please indicate what 'each zone' means in this context. Please also include any exclusion criteria here [i.e., freezing a significant amount of time or not ever going into a social and non-social zone, etc.].

We have specified all zones (non-social and social chambers, intersection/starting point), in the materials and methods section. The plots and statistics of intersection activity (enter frequencies) were also added to Figure 1.

L393: Please include any exclusion criteria here for the Slalom test. Same for L403, L409, L413, and L420.

We did not apply any behaviour-based (freezing) or statistical-based (outliers) exclusion criteria in any test used. Varying sample sizes might stem from videos that we were unable to analyse due to unexpected movement of the apparatus. Since animals were allocated to the test in a previously randomised order, and several animals were assessed simultaneously, the number of animals dropped from each experimental group might differ.

L394: Please state the dimension of your plus-maze apparatus and the height of water-column for the center zone, and the deep and shallow arms.

Manuscript has been updated accordingly.

L410: Please indicate the size of the diameter of the well/arena.

Manuscript has been updated accordingly.

L423 and L432: Please state whether all tested animals were socially reared for Experiment 1 and 2.

In the “Animals” subsection of the Materials and Methods it is clarified that all fish were socially reared except for one experiment (social isolation experiment) where half of the subjects were isolated for 3 days.

L433: Is the 8 referring to 8 dpf here? Also, why were 8dpf zebrafish not tested?

Indeed, 8 refers to 8 dpf. This age group were introduced to the test to represent a zebrafish control without strong sociality.

L437-8: What was the group size of house-mates for social housing?

Each tank (2.5 L) initially contained approximately 40 early-stage larvae. However, due to natural mortality, this number typically declined. At the end of the first month we further reduced stocking densities to 20–25 individuals.

L444 and L451: Please state the age of the zebrafish. Were zf also 30 dpf?

L447: Please state the age of both fish species.

Manuscript has been updated accordingly. In Experiments 4, 5 and 6 zebrafish and paradise fish at the age of 30 dpf were used (lines 557-569).

Small Typos/errors

There are some trivial grammatical mistakes, unnecessary adverbs and colloquial terms, typos, missing commas etc. that make text difficult and informal and/or change the message of the sentences occasionally. Below are some examples, not an exhaustive list. Please ensure these and similar errors are corrected throughout the manuscript.

L23: Intra- and “interest” should be corrected.

L55-57: Use of the future tense “will” should be corrected. Ensure consistent spelling of behaviour throughout the manuscript.

L58 and L279: While not incorrect, please consider using a less colloquial terms instead of 'exactly', 'despite the fact', or removing these. Also, the question should be phrase 'what can we learn' in L58.

L62: 'Currently' should be 'recently'.

L66 and L280: The 'or' should be an 'and' in the listing of behaviours.

L67: Please check spelling; soliters should be 'solitary'.

L108: 'snow' should be 'show'.

L113-6, L277-279, L290-302: Use of the present tense should be corrected to past tense for existing literature and observed results.

L270-1: Spelling of showed.

L288: Missing linking verb 'is'.

Thanks for listing these typos and errors; we have addressed them in the revised version of the manuscript.

Reviewer #2 (Remarks to the Author):

I highly recommend this article for publication. It is a well-executed and original study that underscores the importance of considering the natural history of a species when designing and conducting assays for exploration, anxiety, and working memory in fish. The research compares the sociality of juvenile zebrafish with that of paradise fish, concluding that zebrafish are better suited for social assays, whereas paradise fish are superior models for non-social novelty tests. The study provides compelling evidence supporting the use of paradise fish as a complementary model in biomedical sciences, particularly for individual-subject-based assays of exploration, anxiety, and working memory that require isolation—a significant confounding factor for other mainstream behavioral models.

I have only minor comments, which I believe will improve the article's readability and the replicability of its experiments.

We are extremely grateful to the reviewer for the very positive and very encouraging assessment. We also thank for the constructive suggestions – we have tried to address all the issues that were raised in the revised version of our manuscript.

Discussion

It would be valuable to discuss differences observed in adult fish across tests other than the U-shape. Additionally, consider how the findings of this study could inform future research.

We have thoroughly revised the entire Discussion section to ensure a comprehensive interpretation of our findings. The updated section now includes a detailed discussion of each result, comparison with relevant literature, acknowledgment of the study's limitations, and a brief outline of potential directions for future research.

In lines 23–25, it is mentioned that intra- and inter-test repeatability measures of the anxiety tests revealed that paradise fish express more consistent exploratory and defensive behaviors regarding time and context compared to zebrafish. However, this is not further discussed in the article, despite its importance for cognitive tests and for assessing personality traits.

The relevant sentence of the abstract (lines 23-25) is referring to the results of Experiment 5 aiming to assess behavioural consistency in the two species. The design of this experiment is described in the “Experimental design and analysis” section under the Experiment 5 subheading (lines 562-566), while the results are presented within the Results section, lines 247-273 (section starting with “Paradise fish

show behavioural consistency through time and contexts”). The corresponding figure is Figure 5, and the results are also highlighted in the Discussion section (lines 397-406).

In lines 295–302, it should be emphasized that longer periods of isolation need to be tested in future experiments. Social manipulation outcomes have been highly variable, with no consistent patterns emerging in zebrafish experiments (e.g., Buenhombre et al., 2021).

Thank you for pointing out this important feature of the social isolation protocols and for suggesting a less general interpretation of these results. We concur with this and in the revised version of the text we try to emphasize both in the title of the result section and also in the discussion section of our manuscript that this particular design represents a subchronic treatment for fish. It is true that most social isolation protocols are longer and are applied to younger zebrafish compared to this specific case and those are usually referred to as chronic developmental isolation (see also PMID: 32350040). We argue, however, that this experimental design gives important insights about how differently these two species react to social isolation that takes place prior or during experimentation (e.g. serial sampling through a few days). Our results show that zebrafish immediately react to isolation rearing while paradise fish do not show indications of altered anxiety-like behaviour or locomotion even following three days of this environmental perturbation. Applying chronic developmental isolation to these fish would be also a very interesting direction and we absolutely plan to pursue it in our future work.

In line 303, while noting that paradise fish are more successful in novelty exploration across multiple paradigms, future studies should examine results with larger group sizes, such as a shoal, to enhance generalizability.

Thank you for the suggestion. Indeed, enhancing the group sizes would be crucial in future studies for better generalisability. In this particular setting we aimed to assess the most common ways of designing novelty exploration experiments (using single subjects) and compared it with something which represents social company but still feasible (using two subjects). Since individuals in the same test are not independent from each other, fish tested together have to be considered single biological samples to some extent. This means that enhancing the group sizes would also multiply the required sample sizes several fold. As this study is more exploratory in nature, in our experimental design we aimed to minimise such numbers to comply with the recommendations of our local Animal Welfare board.

Results

It would be helpful to specify the experiment number under each heading to improve clarity.

We apologise for this omission. The experiment numbers have been added to the revised version of the manuscript.

Materials and Methods

The method of video recording is not described. Include details such as the camera used, and environmental conditions (e.g., illumination, background colors, distance from the camera).

We regret this error, and we amended the manuscript to correct it. Experiments were conducted using a Zantiks MWP unit (Zantiks Ltd., Cambridge, UK) with its integrated camera resulting in 640x480 pixels, 30 fps resolution recordings. Recording details are covered in lines 564-565.

Regarding environmental modifications, the composition of the groups is unclear. Specifically, how many animals were housed per group for the social condition prior to experiments 3 and 4? This information is crucial for replication, as stocking density significantly influences behavior (see Buenhombre et al., 2021).

Fish larvae of both species are moved from the incubator to the facility at the age of 5 dpf, where they are kept in 2.5L containers. At the beginning stocking density is 40 fish/tank, at one month post fertilization we reduce the density to 20-25 fish/tank, and around two months of age to 10-12/tank. We have added now this information to the revised manuscript (lines 448-463).

Questions regarding sample sizes are answered in a single comment. All questions are listed below, and the general comment follows.

In lines 359–365, it is unclear how many animals were assigned to isolation and how many to social housing.

Experimental Design

In lines 424–425, it appears that each "N" corresponds to each challenge, but this is not explicitly stated. Clarification is necessary.

In line 433, the context for each "N" is difficult to understand, particularly for paradise fish, where varying sample sizes are reported (e.g., n=21, 19, or 8; n=24, 30). Testing animals in pairs may introduce a confounding factor. For example, although zebrafish are social animals that form shoals, they exhibit increased aggression and dominance when in pairs (see Teles and Oliveira, 2016). This factor should be addressed in the discussion and results for the slalom test.

In line 437, it is unclear how many fish were isolated versus socially housed prior to the experiment. It seems that the first number in parentheses represents the "N" for social housing and the second for isolation, but this should be explicitly stated.

In lines 442–445, the concentrations of buspirone used are not detailed, though they are mentioned earlier. Specify which "N" corresponds to each dosage.

General answer to the questions regarding sample sizes follows (in particular focus on the social isolation and the buspirone experiments):

We apologize about the group size designations being confusing in our original manuscript. In the revised version we clarified which sample size belongs to which group, both in the experimental design and the analysis section.

The final sample sizes for each experiment were influenced by several factors. Some of these factors were biological (i.e. the original number of larvae that were born reached the age of experimentation (27 dpf)), but some were of technical nature (i.e. we aimed for an even allocation to 4 groups in each species (social - three days repeated testing, isolated - three days repeated testing, social - test-naive controls for day2 and social - test-naive controls for day3). A few animals were lost due to rare experimental errors (e.g. subjects were lost due to mispipetting, and some videos were unanalysable due to the movement of the experimental box).

We aimed to allocate at least 20 subjects for each group in each species. In the case of paradise fish it was not possible due to the number of fish at the time of experimentation (~70), hence we allocated approximately 15 to each social group (45 in total) and 25 to the isolated group. The higher N in isolated fish numbers was motivated by the use of multiple social control groups and that isolated fish tend to show bigger variability sometimes. The final numbers (social - three days repeated testing: 13, isolated - three days repeated testing: 23, social - test-naive controls for day2: 13 and social - test-naive controls for day3: 14) were relatively smaller due to the aforementioned issues.

In the case of zebrafish, due to the higher number of animals (~80) we were able to assign 20 to each of the four groups. The final numbers (social - three days repeated testing: 19, isolated - three days

repeated testing: 19, social - test-naive controls for day2: 15 and social - test-naive controls for day3: 18) were relatively smaller due to the aforementioned issues.

We are well aware that such mistakes are sub-optimal, but for a 4 day experiment using approximately 150 animals it is sometimes hard to avoid.

In the case of the buspirone experiment, unfortunately some video recordings got corrupted, and we were unable to involve them in the analysis. This event affected the groups in an uneven manner due to the previous randomisation of subjects during the experiments.

In lines 446–449, it is unclear whether the OT or the SPM experiment was conducted first. The sequence of experiments should be clarified.

The order of tests were OT, inter-test-interval in home tank, then SPM. This is clarified in the revised manuscript.

Reviewer #3 (Remarks to the Author):

This study compares zebrafish and paradise fish juveniles in a series of sociality and novelty coping assays. The authors show that the paradise fish is less affected by presence or absence of social cues and is therefore a better model for studies where social isolation and individual tests are needed. The study is well executed and the paper is well written. But I have several comments and concerns about the assays and interpretation of results. Generally, there is also a major restructuring of sections needed. Please find below my specific comments.

We are grateful for the overall positive assessment of the reviewer, and also for the thoughtful comments and suggestions. We tried to incorporate as many of these as possible in the revised manuscript, to increase clarity and legibility.

Lines 18 solitary instead of solitaire

The typo has been corrected.

Line 23 what is interest repeatability?

This typo has now been corrected to “intertest repeatability”.

Lines 24-25 more consistent...regarding time and context... this is not clear. Please rephrase.

We made an effort to make the phrasing clearer in the abstract (lines 25-27): “Intra- and intertest variability measures of the swimming plus-maze and open tank anxiety tests revealed that, compared to zebrafish, paradise fish express more consistent, repeatable patterns of exploratory and risk-avoidance behaviour across time and contexts.”

Lines 25-26 how can arm alteration test behavioural consistency? This tests flexible learning.

We tried to express here that arm alteration-based exploration is a strategy in which the arms are not chosen randomly during exploration, but based on a strategy to choose a novel, unexplored area and this is expressed in a consistent manner. This requires both working memory and behavioural consistency. However, we understand that this phrasing could be confusing, so for the sake of clarity we changed this in the revised abstract (lines 27-29).

Lines 47-48 is till missing in this model

Line 67 function as solitary individuals.

Both typos have been corrected.

Line 71 with or without the presence of conspecifics?

We tried to rephrase the sentence to increase clarity (line 87-88).

Line 75 zebrafish,

The typo has been corrected.

Lines 74-78 this should be dedicated to hypothesis or predictions, and not the result.

We appreciate the review's observation regarding the structure of the introduction. A clear hypothesis is indeed presented in the introduction (lines 81-82), positioned immediately before the summary of the aims (lines 83-91) and key results (lines 91-93). We acknowledge that introduction styles may vary across disciplines and journals, and that the inclusion of a brief summary of results – commonly associated with the “American style” introduction (see APA style guidelines for articles) – is one such approach. Given the complexity of the study, including the number of research questions, experiments, and findings, we have chosen this structure intentionally. Our aim is to provide readers with a clear roadmap of the manuscript.

Comments on manuscript structure:

Thank you for these suggestions on the structure of the manuscript. Given that there are several suggestions, we collected those below and answered those in a single comment.

- *Lines 85-89 these lines actually belong to the analysis section and refer to what I write above in my comment.*
- *Lines 118 -124 these sentences do not belong to results section. They are methods and analysis.*
- *Lines 141-152 again, these belong to the methods.*
- *Lines 172-175 these belong to the discussion.*
- *Lines 212-224 this whole paragraph belongs to methods section. Also, certain things are redundant as they have already been written in the methods.*
- *Lines 257-259 should go to the discussion.*

We have made a concerted effort to restructure the revised manuscript. Specifically, we have relocated all interpretative content from the Results section to the Discussion section in order to adopt a more traditional structure of scientific writing, reduce the tone of our claims, and avoid potential overinterpretation.

However, we have chosen to retain certain descriptions of experimental procedures in both the Materials and Methods and the Results sections. While this introduces some redundancy, we believe it significantly enhances the clarity and readability of a lengthy manuscript. Additionally, we prefer to include a brief summary sentence highlighting the key findings at the end of each Results subsection, as we find this helps guide the reader through the narrative and emphasizes the main conclusions of each experimental segment.

Line 101 here and everywhere else, the no. of decimal places needs to be constant.

Thanks for the suggestion: we use the standard 3 decimal places in the updated statistical tables in the revised manuscript.

Line 108 show preference towards a conspecific in either condition.

The typo has been corrected.

Lines 112 that zebrafish lost its sociability is not agreeable. The results might be statistically non-significant, but the effect size may be still large. It is clear from the graph 1C that the

zebrafish continue to prefer going to the conspecifics much more than the heterospecific or empty chamber. This is well established from previous research on shoaling preferences in this species, and anything otherwise would be strange. I would advise to consider the effect size for all tests and re-evaluate these results.

There is also a lot of interindividual variation in zebrafish (fig 1C), and there are a lot of fish that did not move or were inert. This kind of indicates that they did not like the setup or were shy or both. This must be accounted for. Or is this because of the darkness in Zantiks experiment chambers? I say this from personal experience in working with Zantiks.

We agree with the Reviewer that the initial phrasing appears to overstate our findings, and we appreciate the opportunity to clarify this in the revised version of the manuscript. Indeed, we do not think that these zebrafish lost their social preference, we just wanted to state that they showed their preference in a less pronounced way. We changed the relevant text accordingly (lines 124-128): “In summary, zebrafish, but not paradise fish, exhibited social approach behaviour in response to a conspecific. Conversely, paradise fish, but not zebrafish, showed avoidance behaviour when exposed to the other species. In the choice condition, zebrafish displayed a trend toward preferring the conspecific, whereas paradise fish showed a greater tendency to approach the heterospecific individual.”

We also agree with the Reviewer that the effect sizes cannot be ignored. Nevertheless, in a condition where individual variability is higher, it takes more observations to obtain the same effect size. Our experimental design, including our choice of sample sizes, was based on our previous experiments (PMID: 32350040), where we applied the original protocol (PMID: 32366356). When a zebrafish has to choose between a conspecific or an empty chamber, the choice appears easier for the animal, the individual variability, therefore, is smaller so the effect size of sociability was measurable. This is a potential factor that could explain the varying results of the statistical tests applied on the same effect sizes. Given the general consensus on reporting data, we cannot overinterpret a non-significant comparison by concluding that there is a difference according to the effect size. However, in designing future experiments which feature similarly complex conditions as multiple influencing factors, we will account for the enhanced variability.

Line 113 what is active aversion from a passive state?

Our intention was to provide an interpretation for our somewhat unexpected observation showing that paradise fish exposed to the choice between an empty and a conspecific-occupied chamber (original protocol) do not show active preference towards the conspecific. We interpret this as those fish being in a passive state regarding the social choice. In contrast, in the inter-species challenge, paradise fish choose an empty chamber over a zebrafish-occupied one - the most straight-forward interpretation of this phenomena is avoidance. In addition, in a double-species challenge, paradise fish spend more time in the proximity of the other species compared to the conspecific. This trend can be either interpreted as a preference for the non-conspecific (zebrafish) or an avoidance from the conspecifics. Since in the inter-specific challenge, we did not measure any preference, but aversion from zebrafish, it is very unlikely that the choice of paradise fish in the double species challenge is motivated by preference. It is more likely that what we are observing is avoidance. Consequently, we interpret the lack of preference in a conspecific challenge as the result of a passive state, while the behaviour in the other two challenges can be seen as active avoidance.

In the manuscript we also provided an alternative explanation for the behaviour observed in the double-species challenge. Briefly, the enhanced approach of the other species compared to the conspecific can be explained with the phenomenon of social buffering. It is possible that the paradise fish only risk the

exploration of the other species, if a conspecific is presented, due to the advantages of being in a social group.

Line 117 is not biased by the presence of

The typo has been corrected.

In figure 2B, units are missing.

In Figure 2B mean exploration latency is shown. This value is derived from scaled time latencies for each chamber, hence it does not have a unit. We provide a description of this variable both in the figure legend and in the “Materials and methods” section.

Line 133 mixed modelling

The typo has been corrected.

Line 138 what is meant here by exploration being more effective?

An effective exploratory behaviour means that the respective group can explore more of the chamber in a standard amount of time compared to the other groups.

In figure 3B, C units are missing.

In Figure 3B (3D in the revised manuscript) mean exploration latency is shown. This value is derived from scaled time latencies for each chamber, hence it does not have a unit. In Figure 3F and G, the unit is secundum - in the revised version of the manuscript we are indicating this, thanks for highlighting the error.

Lines 155 F values should be written as F3,139

All statistical parameters are now included in the new Supplementary Tables.

Lines 167-171, 179-182 it is advisable to have a table instead of listing all the stats etc. here.

We added 7 Supplementary Tables with all statistical parameters in the revised manuscript.

Lines 202 paradise fish is not affected by acute social isolation.

Here we measured different variables indicative of surface-avoidance behaviour, hence we can only state that the surface avoidance of paradise fish is not biased by social isolation. We are sorry for the confusing phrasing - in the revised manuscript we rephrased the respective sentence (line 235) as “in paradise fish it is not biased by acute social isolation”.

Line 222 this should include inter and intraindividual variability. Is that considered?

Indeed, both types of variabilities have been considered. Within-test variability represents interindividual variability in the same test type, while between-test variability represents intraindividual variability between different test occasions.

Line 225 this is not clear, please rephrase.

Thanks for highlighting this. We have rephrased this sentence to avoid confusion. In the revised version of the manuscript (lines 260-261) now we state: “Swimming velocity and the number of immobile episodes showed big within-test variability and small inter-test variability, consequently these readouts are the most repeatable in both species”.

Lines 224-232 the authors must include a table with all these results, within and between test variability, R measures, CI and P values. It is hard to understand these results in the way they have been presented at the moment. Similar holds for the correlation tests.

We agree with the reviewer that the original version of presenting the result of the statistical tests was confusing. For clarity, in the revised manuscripts all the statistical results and related values have been moved to the Supplementary Table 5.

Figure 5D estimates of repeatability, whether significant or not, would decide if the behaviour is significantly repeatable. At the moment, it is hard to understand, give that these estimates have not been tabulated. A non-overlap of the bars would imply that the R estimates are significantly different from each other.

On Figure 5D we show repeatability scores using different geometrical shapes corresponding to each variable type and the confidence intervals. As the Reviewer 3 assumed, if the confidence interval does not intersect the 0 line on the plot, we can consider that a significantly repeatable measure. Also true, that non-overlapping confidence intervals are the indication of significant difference in repeatability of measures. We clarify this in the current figure legend for Figure 5, and the revised manuscript also contains a table with the aforementioned parameters (Supplementary Table 5).

Lines 247-251 these belong to the methods. Moreover, can you confirm if the hypothesis is that 'fish that make more alterations will be considered to have better working memory'?

Lines 314-319 as mentioned above, and as reiterated by the authors here and in the next paragraph, the y maze movement pattern tests for exploration and not working memory. I have concerns about it being used to assess working memory. The fact that zebrafish use more indirect visits probably show indicate that it is more random in its behaviour.

See also in “comment on manuscript structure”.

Our null hypothesis (H0) here is that the exploration strategy and the working memory of the two species are similar. Alternatively, (H1) they show similar exploration strategy but with different efficacy (different working memory), or (H2) they show different exploration strategy, therefore the working memory is hard to compare. If both species explore the y-maze just as rodents do, the arm alternations will be indicative of working memory. Animals using this strategy would show alternation frequencies between 50% (random) and 100% (complete alternations). In zebrafish, however, alteration frequencies are below 50%, which suggests that this species follows a different, non-random strategy, compared to paradise fish, mice or rats. If this is the case, we cannot use this test in its original form to measure working memory in juvenile zebrafish, whereas the test would work just fine to assay working memory in juvenile paradise fish.

Nevertheless, if the two species are using fundamentally different exploratory strategies, a “worse” performance of juvenile zebrafish in the y-maze test does not indicate a worse working memory. Indeed, the innate differences in the exploratory behaviour make such comparisons impossible. As we state in the manuscript (lines 408-412): “In our study, paradise fish and zebrafish showed approximately 70% and 40% alternations, respectively. This suggests that paradise fish exhibit functional working memory, whereas zebrafish performance falls below the threshold of random choice, indicating distinct exploratory strategies between the species. Forty percent alternation is also supported by the free-movement pattern y-maze test usually applied in zebrafish.” We can only conclude that zebrafish show a different exploration strategy, which is less efficient for exploring such an arena. This alternate exploratory behaviour might also require less working memory, but this cannot be concluded from our experiments.

Lines 251-255 The velocity differences are not talked about here.

Generally, all this should be presented as a table. I would advise summarizing the results of all the behavioural tests in a comprehensive table.

Thank you for pointing out that we did not report the differences in velocity, we corrected it in the updated manuscript. In the revised version (lines 296-297) we state: “Interestingly, higher alternation % is accompanied by lower swimming velocity in paradise fish compared to zebrafish.”

Line 256 which exceed alterations of zebrafish (x %).

This has been corrected in the revised manuscript (line 295).

Figure 6C legend and x axis labelling present redundant information. Further, the observation that zebrafish had higher velocity despite making less visits, revisits and alterations indicate that the fish could be stressed and thus made random movements.

Thank you for this important suggestion. We also considered this possibility, however, the alternation percentage of zebrafish is below 50% hence we cannot consider their choices random. We added this argument to the Discussion part of the revised manuscript.

Lines 270-271 show and not showe

The typo has been corrected.

Lines 275-276 the figure description is insufficient to explain how effect sizes contribute and what the figure means. A detailed description is needed. Anxilytic should be replaced by anxiolytic.

Thank you for pointing out the need for clarification in the figure legend. We updated the figure with a more detailed description of the calculations of every parameter on the plot. The typo in the figure has been also corrected.

Lines 277-279 adult and juveniles are ought to differ in behaviour. And the explanation here belongs to introduction where the authors must explain why they did this study on juveniles rather than on adults.

The revised version of the manuscript was updated to specify why we think that larval and juvenile stages of these fish are advantages compared to other model organisms (e.g. lines 85-87). Briefly, high-throughput screening of behaviour due to the number, compactness and short generation time of animals, as well as whole-brain imaging due to the transparent skull of young fish is only available at these stages.

Lines 290-293 this can be instead interpreted as social inquisitiveness, with paradise fish making more inspections than zebrafish.

Social inquisitiveness or curiosity and anxiety are both triggered by novelty and emerge in the same situation. This can cause an approach-avoidance conflict in a wide range of species including fish and humans (PMIDs: 13252152, 17981434, 29568703, 39414697). Most of the anxiety tests currently in use are based on anxiety-like responses triggered by novel stimuli (PMID: 22981935).

In the manuscript we state that curiosity can be a motivation for paradise fish to explore non-conspecific zebrafish, which only happens in the presence of a conspecific. In other words, under normal circumstances curiosity might be inhibited by anxiety-like states, but this might be ameliorated by the presence of the conspecific, a phenomenon known as social buffering (PMID: 36285460).

Lines 294-295 there is no predator here and so I doubt this can be related to that. The zebrafish seem to be smaller in size than the paradise fish and so it cannot be a predator.

We apologise for overinterpreting the results of the referenced study. In the manuscript we wanted to provide examples of social buffering in different species to support our related argument about social

buffering. For clarification we added the following paragraph to the Discussion section (lines 348-352) in the revised manuscript: “Social buffering of anti-predator behaviour has previously been observed in late larval-stage paradise fish, as well as across diverse fish species in response to stressors ranging from novelty to predation cues. These effects have been documented in solitary individuals, loosely aggregating species, and those with defined social structures, suggesting that social buffering is a conserved mechanism with a general stress-ameliorating function across fish taxa”. While the aforementioned study in paradise fish assessed social buffering specifically in the presence of predator cues, it also supports that the background mechanisms of the social buffering is apparent in general in this species. Based on the wide range of situations where social buffering can arise in multiple species, we assume that this mechanism has a general stress-meliorating effect through different challenges regardless of the certainty of the threat. Social buffering has been documented in response to a range of cues from novelty to different predator signals in a wide range of vertebrates, including several fish species living either alone (PMID: 32452106) or in loose aggregations (PMIDs: 28361887, 28361887) or in well-defined social structures (PMIDs: 31506060, 23628383).

Line 302 presence or absence of conspecifics

Typo has been corrected in the revised manuscript.

Lines 326-328 this sentence is not clear. What is a R strategist?

The “r-strategist” (or “r-selected species”) designation in ecology refers to species that are characterised by small size, quick reproduction time and short lifespan due to their fluctuating environment (see Jeschke 2008 <https://doi.org/10.1016/B978-0-12-409548-9.11121-2>). Zebrafish are generally described as opportunist animals whose quick reproduction cycle is associated with raining and the associated food surplus, hence they can be considered r-strategists. These species cope with environmental challenges with adaptation on the population level (producing a new generation of individuals in great numbers) instead of adaptation on the individual level (phenotypic plasticity). R-strategists are less likely to give plastic responses, as they rely on their innate behavioural repertoire.

We acknowledge, however, that using this term in the given context is too speculative and would require a more in-depth explanation. As such an argument is out of the scope of our manuscript, we removed it from the revised version.

Lines 332 the juvenile zebrafish and paradise fish are different in body size. I think this difference itself can contribute to differences in behaviour. The authors must discuss this.

Thank you for highlighting this important potential confounding factor in our research design. We agree that physical differences such as body size can influence and in some cases limit behaviour. To investigate this as a potential interfering factor, we measured body length on the video recordings of the social isolation experiment of both species. We choose this experiment to make the measurements because social isolation can also bias development and body size. We added the body length and also centrum entries (locomotion) data to the respective part of the Result section (lines 187-190) and to the relevant figure (Figure 3B and 3C). Briefly, we found that juvenile paradise fish are shorter compared to zebrafish of the same age, and this is independent of isolation or social rearing at the investigated period. We also found in all of our experiments that paradise fish swim slower and less distance compared to zebrafish in social and non-social challenges, in anxiety tests or in the y-maze. However, paradise fish consistently explore areas more efficiently: they visit more chambers of the slalom test and make more alternations in the y-maze compared to zebrafish. In the case of the slalom test, only paradise fish reached close to 100% efficacy (visited all 12 chambers). Based on these it is obvious that the smaller body size and less locomotion do not prevent paradise fish from exploring their environment efficiently. We also want to note that apart from the y-maze test, we never compared paradise fish and

zebrafish directly, but made comparisons to their own control groups and reported if a factor (like social company or treatment with a small molecular reagent) has an effect on the species behaviour. We prefer these comparisons in cases where the behaviour is not limited by locomotion, as it rules out the potential interference of locomotive differences between the two species.

In the revised version of the manuscript we now discuss this potential confounding factor (lines 386-396.)

Lines 348-349 zebrafish between 10 and 30 days are not just juveniles. The are larvae around till 20-25 days beyond which they become juveniles. Did you consider that change?

Thank you for highlighting the erroneous use of the accepted nomenclature on our part. In the revised version of the manuscript we tried to correct this, using the accepted staging described in the Zebrafish book (https://zfin.org/zf_info/zfbook/stages/). Indeed, the observed period spans between the typical larval stage (10 dpf) and early juvenile stage (30 dpf). This period represents a less investigated period for zebrafish behaviour. However, a morphological-centred staging nomenclature is not sensitive to account for important behavioural transitions that could reflect the maturation of the neural circuits essential for behaviour. For example, while the morphological juvenile stage starts around 30 dpf, an important behavioural metamorphosis occurs much earlier, around 14 dpf in zebrafish (PMIDs: 21262817, 26347614, 30410116, 32350040).

In our study, we wanted to observe fish during this transition both for social and for avoidance behaviours. We also note that the behavioural repertoire during this period is much expanded compared to the typical larval stage in the literature (~5-8 dpf). We argue, therefore, that the post 14 dpf period should be considered as a distinct “larval” stage because, due to the appearance of novel behavioural elements. Incidentally, this period also represents an important window of opportunity when, similarly to larvae, high-throughput screening of behaviour and in vivo whole-brain imaging is still available, yet the social, cognitive and affective behaviour of the animals is already present in a relatively mature form.

Line 354 terminated with tricaine?

We have clarified that we used tricaine overdose for euthanasia in accordance with current Hungarian legislation (40/2013. (II. 14.) Government decree).

Line 384 why did you use larval and juvenile fish? This is not clear and should be explained in the introduction. Larvae and juveniles can be differently active, are much less social than adults and this can be a potential confound. For example, see studies by Roy and Arlinghaus 2022, Sbragaglia et al. 2021, 2022 where they measured ontogenetic change in collective behaviour in zebrafish.

We specifically focused on this period as it represents a window of opportunity when we can exploit many of the advantages of the early larval stage and some of those of the adult stage. Zebrafish larvae are highly accessible for manipulation and we can also monitor their physiological and behavioural features. However, pre-metamorphosis these measurements can be very variable, and as many (larval) individuals do not even reach the juvenile stage (PMID: 23811824), arguably, it is less representative to the species. On the other hand, the logistics of testing fully grown, adult animals in these paradigms is comparable to running these tests in mice, hence we would lose one of the main advantages of this model species. The observed period, where we combine the typical larval (10 dpf) and early juvenile stages (30 dpf), we believe, is where the use of high-throughput methodologies, pioneered in fish, can be the most informative.

Line 414 the working memory test is not clear. How do you test working memory actually in these fish? If it is using the FYMP, then I am not sure how much learning and memory is involved here. It has a strong confound with fish exploratory behaviour, and alteration or direct or indirect t revisits can be easily interpreted as exploration. I am aware that there are previous studies in mice that have used this since the method is developed using the Zantiks. But I would be highly conservative to regard this as working memory and not as mere exploratory behaviour. And the results resonate my observation above (ref to lines 255-259 in the main text).

To test working memory and exploration strategies we used a y-maze apparatus similar to that used for mice and rats. Our apparatus was indeed similar to the FMP-Y maze developed by Zantiks (PMID: 32748238), however, the analysis was based on the rodent version (see method description in PMID: 30535683). As explained in the “Results” section (lines 292-294) and in the respective subsection of the “Materials and methods” (lines 521-523), alternation was defined as the fish entering all three arms consecutively. We calculated this value in an overlapping manner: for example, a pattern of A–B–C–A would consist two alternations (A-B-C and B-C-A). We amended the respective figure legend as well the Methods section in the revised manuscript (lines 528-532). We note that the approximately 40% of alternations observed in zebrafish seems to be in line with the data reported by Parker (PMID: 32748238) using the FMP-Y maze (in this paper, however, the authors define alternation as switching between right and left turns).

Line 446 why did you have a separate test for behavioural consistency? It could have been tested with the tests that you did previously like open tank, surface avoidance, exploratory test, etc.

The behavioural consistency experiment was one of the first ones to be conducted during our work. At this time we only worked with the open tank (OT) and SPM tests (PMID: 30410116) to analyse larval exploration. Once these experiments were already completed ZKV developed and validated the showjump and the slalom tests, so we were able to combine these with the very sensitive SPM and leave out the less sensitive OT test in later analyses.

Line 447 ‘used’ instead of ‘applied’ zebrafish.

The sentence now has been corrected as suggested.

Line 458 stats was done using R (ref.).

The sentence now has been corrected as suggested.

Lines 457-478 you write what tests were done. But you should describe the analysis by saying what comparisons you did, what were the different variables used, what was the formula for each kind of test, what kind of transformation was done for the data, etc.

In the revised version of the manuscript, all statistical data is available in Supplementary Tables 1-7, describing all comparisons, hypotheses testing method, sample sizes, etc.